# RILe: Reinforced Imitation Learning

## Abstract

Reinforcement Learning has achieved significant success in generating complex behavior but often requires extensive reward function engineering. Adversarial variants of Imitation Learning and Inverse Reinforcement Learning offer an alternative by learning policies from expert demonstrations via a discriminator. However, these methods struggle in complex tasks where randomly sampling expert-like behaviors is challenging. This limitation stems from their reliance on policy-agnostic discriminators, which provide insufficient guidance for agent improvement, especially as task complexity increases and expert behavior becomes more distinct. We introduce RILe (Reinforced Imitation Learning environment), a novel trainer-student system that learns a dynamic reward function based on the student's performance and alignment with expert demonstrations. In RILe, the student learns an action policy while the trainer, using reinforcement learning, continuously updates itself via the discriminator's feedback to optimize the alignment between the student and the expert. The trainer optimizes for long-term cumulative rewards from the discriminator, enabling it to provide nuanced feedback that accounts for the complexity of the task and the student's current capabilities. This approach allows for greater exploration of agent actions by providing graduated feedback rather than binary expert/non-expert classifications. By reducing dependence on policy-agnostic discriminators, RILe enables better performance in complex settings where traditional methods falter, outperforming existing methods by 2x in complex simulated robot-locomotion tasks.

## 1 Introduction

Reinforcement Learning (RL) has emerged as a powerful framework for teaching agents to perform complex tasks. In recent years, deep reinforcement learning has demonstrated remarkable success in replicating sophisticated behaviors, including playing Atari games, chess, and Go (Mnih et al., 2013; Silver et al., 2018). However, these achievements often come at a cost: the tedious and challenging process of designing reward functions, as predicting the policy outcome from a manually crafted reward function remains notoriously difficult.

To overcome the reward engineering problem, Imitation Learning (IL) leverages expert demonstrations to learn a policy. Since vast amounts of expert data are required to accurately learn expert behaviors, Adversarial Imitation Learning (AIL) approaches, such as GAIL (Ho & Ermon, 2016), have been proposed as data-efficient alternatives. AIL employs a discriminator to measure similarity between learned behavior and expert behavior, rewarding the agent accordingly. While computationally efficient, AIL methods suffer from a critical limitation: the policy-agnostic nature of their discriminators. The discriminator lacks any inherent incentive to guide the agent towards expert-like behavior, in contrast to engineered reward functions in RL. Consequently, AIL methods face challenges in complex tasks requiring extensive exploration to find optimal actions. For instance, in digital locomotion tasks, AIL methods often struggle to consistently replicate expert performance (Peng et al., 2018).

Inverse Reinforcement Learning (IRL) is another approach to alleviate reward engineering. Unlike IL, which directly learns expert behavior, IRL seeks to infer the underlying reward function that motivates the agent to acquire expert behaviors. The reward function and the agent are trained iteratively, with updates to the reward function based on the agent's behavior. This iterative process renders IRL computationally expensive (Zheng et al., 2022). Adversarial Inverse Reinforcement Learning (AIRL) (Fu et al., 2018) attempts to address this inefficiency by introducing a discriminator

that enables simultaneous learning of the policy and reward function. However, in AIRL, the reward function is tightly coupled to the discriminator, potentially limiting its ability to capture complex task structures or long-term dependencies and inheriting the limitations of a policy-agnostic discriminators. This highlights the need for a method that can learn a more flexible reward function without the computational overhead of traditional IRL methods.

To overcome these challenges and effectively learn behaviors in complex settings, we propose Reinforced Imitation Learning (RILe) (Fig. 1-(d)). RILe aims to combine the ability to learn a reward function that actively guides the agent to imitate expert behavior with the computational efficiency of adversarial frameworks. At the core of RILe is a novel trainer-student system designed to address the shortcomings of existing methods:

- A student agent that learns to replicate the expert's policy via RL in the environment
- A trainer agent that learns a reward function via RL and guides the student agent during training

By integrating the trainer-student dynamic, RILe decouples reward learning from policy learning and the discriminator, allowing each component to specialize and thereby overcome the limitations of policy-agnostic discriminators. While RILe utilizes a discriminator similar to those in adversarial frameworks, its role is fundamentally redefined. In RILe, the discriminator's primary function is to provide feedback to the trainer agent by distinguishing expert data from student roll-outs. This feedback serves as the reward signal for the trainer, not directly influencing the student agent. The trainer leverages the discriminator's feedback to learn a reward function that effectively guides the student agent. This approach enables more nuanced reward shaping, particularly beneficial in tasks requiring complex decision-making and extensive exploration.

Our contributions are two-fold:

1. Decoupled Reward-function Learning: We introduce a novel approach where the trainer agent learns the reward function independently from both the student agent and the discriminator. Unlike existing methods that derive rewards directly from discriminator outputs, our trainer agent uses reinforcement learning to optimize the reward function based on the feedback from the discriminator. By focusing on long-term reward maximization, RL enables the trainer to distill inconsistent feedback from the discriminator into meaningful rewards, leading to better student performance.
2. Dynamic Reward Customization: Our trainer agent dynamically adjusts rewards based on the student agent's progress, facilitating a better learning experience and enabling accurate imitation of expert behavior in complex settings. This adaptive approach allows for more gradual learning, particularly in tasks where the optimal behavior may change depending on the agent's current capabilities.

We evaluate RILe against state-of-the-art methods in AIL, and AIRL, specifically GAIL Ho & Ermon (2016) AIRL Fu et al. (2018), GAIfO Torabi et al. (2018b), BCO Torabi et al. (2018a), IQ-Learn Garg et al. (2021) and DRAIL Lai et al. (2024). Our experiments span three scenarios: (1) Tailoring a reward function dynamically in a discrete maze task, (2) Investigating the impact of expert data on the trainer-student dynamics in a humanoid locomotion task, and (3) Imitating expert data in continuous control tasks. The results demonstrate RILe's superior performance, especially in complex tasks, and its ability to learn an effective dynamic reward function where baseline methods fail.

## 2 RELATED WORK

We review literature on learning from expert demonstrations, focusing on Imitation Learning (IL) and Inverse Reinforcement Learning (IRL), which form the conceptual foundation of RILe.

**Imitation Learning**   Early work introduced Behavioral Cloning (BC) (Bain & Sammut, 1995), which learns a policy congruent with expert demonstrations through supervised learning. DAgger (Ross et al., 2011) introduces data aggregation. GAIL (Ho & Ermon, 2016) introduces adversarial methods, where a discriminator aims to discriminate expert demonstrations, while a generator tries to fool the discriminator. BCO (Torabi et al., 2018a) extends BC and GAIfO (Torabi et al., 2018b) extends GAIL to state-only observation scenarios. DQfD (Hester et al., 2018) proposes two-stage approach with pre-training, and ValueDice (Kostrikov et al., 2020) uses a distribution-matching

objective between policy and expert. DRAIL (Lai et al., 2024) enhances adversarial imitation learning via a diffusion-based discriminator, which improves learning efficiency. Despite progress, IL faces challenges in efficacy and generalization (Zheng et al., 2022; Toyer et al., 2020). RILe addresses these by introducing an adaptive teacher agent to guide the student beyond expert demonstrations.

**Inverse Reinforcement Learning** IRL, introduced by Ng & Russell (2000), learns the expert's intrinsic reward function. Key developments include Apprenticeship Learning (Abbeel & Ng, 2004), Maximum Entropy IRL (Ziebart et al., 2008), and adversarial approaches like AIRL (Fu et al., 2018). IQ-Learn (Garg et al., 2021) reformulates IRL integrates inverse learning of the reward function into Q-learning. Recent work explores handling unstructured data (Chen et al., 2021) and cross-embodiment scenarios (Zakka et al., 2022). Despite advancements, IRL faces challenges in computational efficiency and scalability (Arora & Doshi, 2021). RILe addresses these by jointly learning policy and reward function in a single process.

## 3 BACKGROUND

### 3.1 MARKOV DECISION PROCESS

A standard Markov Decision Process (MDP) is defined by $(S, A, R, T, K, \gamma)$. $S$ is the state space consisting of all possible environment states $s$, and $A$ is action space containing all possible environment actions $a$. $R = R(s, a) : S \times A \to \mathbb{R}$ is the reward function. $T = \{P(\cdot|s, a)\}$ is the transition dynamics where $P(\cdot|s, a)$ is an unknown state state transition probability function upon taking action $a \in A$ in state $s \in S$. $K(s)$ is the initial state distribution, i.e., $s_0 \sim K(s)$ and $\gamma$ is the discount factor. The policy $\pi = \pi(a|s) : S \to A$ is a mapping from states to actions. In this work, we consider $\gamma$-discounted infinite horizon settings. Following Ho & Ermon (2016), expectation with respect to the policy $\pi \in \Pi$ refers to the expectation when actions are sampled from $\pi(s)$: $\mathbb{E}_\pi[R(s, a)] \triangleq \mathbb{E}_\pi[\sum_{t=0}^{\infty} \gamma^t R(s_t, a_t)]$, where $s_0$ is sampled from an initial state distribution $K(s)$, $a_t$ is given by $\pi(\cdot|s_t)$ and $s_{t+1}$ is determined by the unknown transition model as $P(\cdot|s_t, a_t)$. The unknown reward function $R(s, a)$ generates a reward given a state-action pair $(s, a)$. We consider a setting where $R = R(s, a)$ is parameterized by $\theta$ as $R_\theta(s, a) \in \mathbb{R}$ (Finn et al., 2016).

Our work considers an imitation learning problem from expert trajectories, consisting of states $s$ and actions $a$. The set of expert trajectories $\tau_E$ are sampled from an expert policy $\pi_E \in \Pi$, where $\Pi$ is the set of all possible policies. We assume that we have access to $m$ expert trajectories, all of which have $n$ time-steps, $\tau_E = \{(s_0^i, a_0^i), (s_1^i, a_1^i), \ldots, (s_n^i, a_n^i)\}_{i=1}^m$.

### 3.2 REINFORCEMENT LEARNING (RL)

Reinforcement learning seeks to find an optimal policy, $\pi^*$, that maximizes the discounted cumulative reward given from the reward function $R = R(s, a)$ (Fig. 1-(a)). In this work, we incorporate entropy regularization using the $\gamma$-discounted casual entropy function $H(\pi) = \mathbb{E}_\pi[-\log \pi(a|s)]$ (Ho & Ermon, 2016; Bloem & Bambos, 2014). The RL problem with a parameterized reward function and entropy regularization is defined as

$$\text{RL}(R_\theta(s, a)) = \pi^* = \arg\max_\pi \mathbb{E}_\pi[R_\theta(s, a)] + H(\pi). \tag{1}$$

### 3.3 INVERSE REINFORCEMENT LEARNING (IRL)

Given sample trajectories $\tau_E$ from an optimal expert policy $\pi_E$, inverse reinforcement learning aims to recover a reward function $R_\theta^*(s, a)$ that maximally rewards the expert's behavior (Fig. 1-(b)). Formally, IRL seeks a reward function, $R_\theta^*(s, a)$, satisfying: $\mathbb{E}_{\pi_E}[\sum_{t=0}^{\infty} \gamma^t R_\theta^*(s_t, a_t)] \geq \mathbb{E}_\pi[\sum_{t=0}^{\infty} \gamma^t R_\theta^*(s_t, a_t) + H(\pi)] \quad \forall \pi$. Optimizing this reward function with reinforcement learning yields a policy that replicates expert behavior: $\text{RL}(R_\theta^*(s, a)) = \pi^*$. Since only the expert's trajectories are observed, expectations over $\pi_E$ are estimated from samples in $\tau_E$. Incorporating entropy regularization $H(\pi)$, maximum causal entropy inverse reinforcement learning (Ziebart et al., 2008) is defined as

$$\text{IRL}(\tau_E) = \arg\max_{R_\theta(s,a) \in \mathbb{R}} \left( \mathbb{E}_{s,a \in \tau_E}[R_\theta(s, a)] - \max_\pi \left( \mathbb{E}_\pi[R_\theta(s, a)] + H(\pi) \right) \right). \tag{2}$$

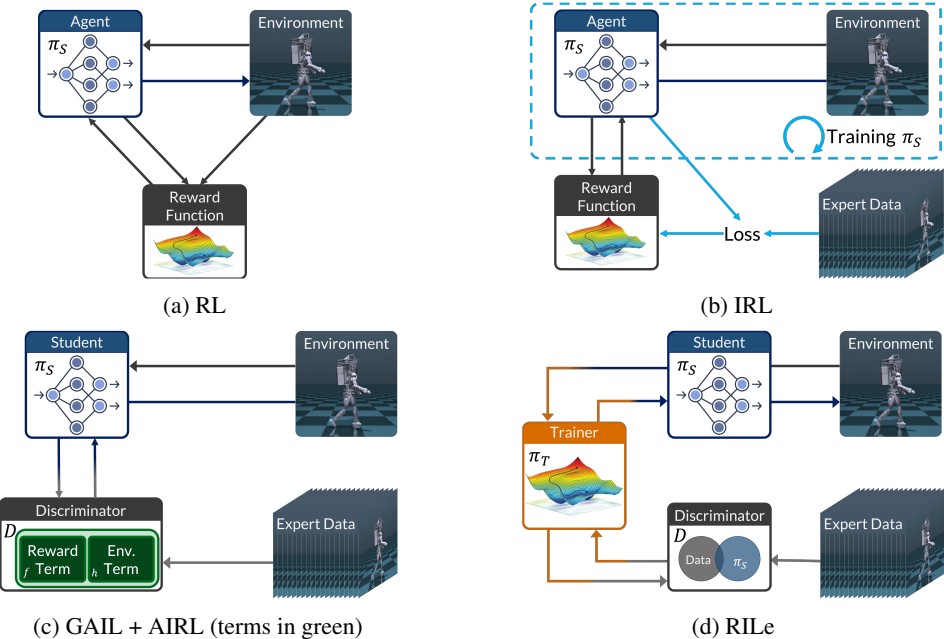

(a) RL

(b) IRL

(c) GAIL + AIRL (terms in green)

(d) RILe

Figure 1: **Overview of the related works. (a) Reinforcement Learning (RL):** learning a policy that maximizes hand-defined reward function; **(b) Inverse RL (IRL):** learning a reward function from data. IRL has two stages: 1. training a policy with frozen reward function, and 2. updating the reward function by comparing the converged policy with data. These stages repeated several times; **(C) Generative Adversarial Imitation Learning (GAIL) + Adversarial IRL (AIRL):** using discriminator as a reward function. GAIL trains both policy and the discriminator at the same time. AIRL implements a new structure on the discriminator, seperating reward from environment dynamics by using two networks under the discriminator (see additional terms in green). **(D) RILe:** similar to IRL, learning a reward function from data. RILe learns the reward function at the same time with the policy, using discriminator as a guide for learning the reward.

### 3.4 ADVERSARIAL IMITATION LEARNING (AIL) AND ADVERSARIAL INVERSE REINFORCEMENT LEARNING (AIRL)

Imitation Learning (IL) aims to directly approximate the expert policy from given expert trajectory samples $\tau_E$. It can be formulated as $\text{IL}(\tau_E) = \arg\min_\pi \mathbb{E}_{(s,a)\sim\tau_E}[L(\pi(\cdot|s), a)]$, where $L$ is a loss function, that captures the difference between policy and expert data.

GAIL (Ho & Ermon, 2016) introduces an adversarial imitation learning setting by quantifying the difference between the agent and the expert with a discriminator $D_\phi(s, a)$, parameterized by $\phi$ (Fig. 1-(c)). The discriminator distinguishes between between expert-generated state-action pairs $(s, a) \sim \tau_E$ and non-expert ones $(s, a) \notin \tau_E$. The goal of GAIL is to find the optimal policy that fools the discriminator while maximizing an entropy constraint. The optimization is formulated as a zero-sum game between the discriminator $D_\phi(s, a)$ and the policy $\pi$:

$$\min_\pi \max_\phi \mathbb{E}_\pi[\log D_\phi(s, a)] + \mathbb{E}_{\tau_E}[\log(1 - D_\phi(s, a))] - \lambda H(\pi). \tag{3}$$

In other words, the reward function that is maximized by the policy is defined as a similarity function, expressed as $R(s, a) = -\log(D_\phi(s, a))$.

AIRL (Fu et al., 2018) extends AIL to inverse reinforcement learning, aiming to recover a reward function decoupled from environment dynamics (Fig. 1-(c)). AIRL structures the discriminator as:

$$D_{\phi,\psi}(s, a, s') = \frac{\exp(f_\phi(s, a, s'))}{\exp(f_\phi(s, a, s')) + \pi(a|s)}, \tag{4}$$

where $f_\phi(s, a, s') = r_\psi(s, a) + \gamma V_\phi(s') - V_\phi(s)$. Here, $r_\psi(s, a)$ represents the learned reward function that is decoupled from the environment dynamics, $\gamma V_\phi(s') - V_\phi(s)$. The AIRL optimization

problem is formulated equivalently to GAIL (see Eqn. 3). The reward function $r_\psi(s, a)$ is learned through minimizing the cross-entropy loss inherent in this adversarial setup. Therefore, the reward function remains tightly coupled with the discriminator's learning process.

# 4 RILe: Reinforced Imitation Learning

We propose Reinforced Imitation Learning (RILe) to learn the reward function and acquire a policy that emulates expert-like behavior simultaneously in one learning process. Our RILe framework introduces a novel trainer-student dynamic to overcome limitations in existing imitation learning methods. Figure 2 illustrates our approach.

In RILe, the student agent learns an action policy by interacting with the environment, while the trainer agent learns a reward function that effectively guides the student toward expert-like behavior. Both agents are trained simultaneously via reinforcement learning, with assistance from an adversarial discriminator.

Unlike traditional AIL, where the discriminator directly influences the student, RILe decouples this process by introducing the trainer agent. The discriminator provides immediate feedback solely to the trainer agent. This decoupling allows the trainer to adjust the reward function on-the-fly considering the current stage of the student's learning process, and guiding the student without waiting for its policy to converge, a significant efficiency improvement over traditional IRL.

In our framework, the trainer agent takes the key role. Trained via RL, the trainer learns to provide tailored feedback to the student by maximizing the cumulative rewards it receives from the discriminator. This approach equips RILe with three key advantages that set it apart from existing AIL frameworks: (1) the trainer associates its reward signals to future improvements in the student's behavior, even if these improvements occur after many steps, (2) the trainer encourages the student to explore actions that steer it in the right direction, even when immediate expert-like behavior isn't achieved yet, and (3) the trainer adjusts its reward function based on the student's current policy, creating a learning path that gradually guides the student toward expert behavior.

This approach enables RILe to overcome limitations of previous methods, particularly in complex tasks requiring extensive exploration, by promoting the discovery of expert-like strategies even when the student's initial policy significantly diverges from expert behavior.

In the following, we define the components of RILe and explain how they can efficiently learn behavior from imperfect data.

**Student Agent** The student agent learns a policy $\pi_S$ by interacting with an environment in a standard RL setting within an MDP. For each of its actions $a^S \in A$, the environment returns a new state $s^S \in S$. However, rather than from a hand-crafted reward function, the student agent receives its reward from the policy of the trainer agent $\pi_T$. Therefore, the reward function is represented by the trainer policy. Thus, the student agent is guided by the actions of the trainer agent, i.e., the action of the trainer is the reward of the student: $r^S = \pi_T((s^S, a^S))$. The optimization problem of the student agent is then defined as

$$\min_{\pi_S} -\mathbb{E}_{(s^S, a^S) \sim \pi_S}[\pi_T\left((s^S, a^S)\right)]. \tag{5}$$

**Discriminator** The discriminator differentiates between expert-generated state-action pairs $(s, a) \sim \tau_E$ and state-action pairs from the student $(s, a) \sim \pi_S$. In RILe, the discriminator is defined as a feed-forward deep neural network, parameterized by $\phi$. Hence, the optimization problem is

$$\max_{\phi} \mathbb{E}_{(s,a) \sim \tau_E}[\log(D_\phi(s, a))] + \mathbb{E}_{(s,a) \sim \pi_S}[\log(1 - D_\phi(s, a))]. \tag{6}$$

To provide effective guidance, the discriminator needs to accurately distinguish whether a given state-action pair originates from the expert distribution $(s, a) \sim \tau_E$ or not $(s, a) \notin \tau_E$. The feasibility of this discrimination has been demonstrated by GAIL (Ho & Ermon, 2016). The according lemma and proof are presented in the Appendix B.

**Trainer Agent** The trainer agent guides the student to imitate expert behavior by operating as its reward mechanism. Because the trainer cannot directly observe the student's policy $\pi_S$,

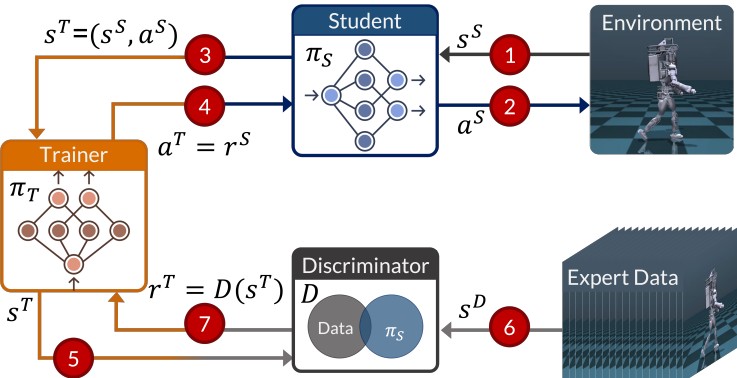

Figure 2: **Reinforced Imitation Learning (RILe)**. The framework consists of three key components: a student agent, a trainer agent, and a discriminator. The student agent learns a policy $\pi_S$ by interacting with an environment, and the trainer agent learns a reward function as a policy $\pi_T$. (1) The student receives the environment state $s^S$. (2) The student takes an action $a^S$, forwards it to the environment which is updated based on $a^S$. (3) The student forwards its state and action to the trainer, whose state is $s^T = (s^S, a^S)$. (4) Trainer, $\pi_T$, evaluates the state action pair of the student agent $s^T = (s^S, a^S)$ and chooses an action $a^T$ that then becomes the reward of the student agent $a^T = r^S$. (5) The trainer agent forwards the $s^T = (s^S, a^S)$ to the discriminator. (6) Discriminator compares student state-action pair with expert demonstrations $(s^D)$. (7) Discriminator gives reward to the trainer, based on the similarity between student- and expert-behavior.

we model the trainer's environment as a Partially Observable MDP (POMDP): $\text{POMDP}_T = (S_T, A_T, \Omega_T, T_T, O_T, R_T, \gamma)$. The state space $S_T = S \times A \times \pi_S$ includes all possible state-action pairs from the standard MDP and the student's policy $\pi_S$, which is hidden from the trainer, introducing partial observability. $A_T$ is the action space, a mapping from $S_T \to \mathbb{R}$, so the action is a scalar value. The observation space $\Omega_T = S \times A$ consists of the observable state-action pairs of the student. The transition dynamics $T_T$ and the observation function $O_T$ are defined formally in Appendix A. The reward function $R_T(s^T, a^T)$ evaluates the effectiveness of the trainer's action in guiding the student, where $s^T = (s^S, a^S)$ is the observation of the trainer. $\gamma$ is the discount factor.

The trainer agent learns a policy $\pi_T$ that produces adequate reward signals to guide the student agent, by learning in a standard RL setting, within $\text{POMDP}_T$. The trainer operates under partial observability and observes the student's state-action pair $s^T = (s^S, a^S) \in S \times A$, without observing $\pi_S$. It generates a scalar action $a^T$, bounded between $-1$ and $1$, which is given to the student agent as the reward $r^S$. If the trainer's reward depends only on the discriminator's output, the trainer receives the same reward regardless of its action, offering no immediate feedback on whether rewarding or penalizing the student was effective. For example, when the student behaves like the expert (discriminator output is $\sim$1), the trainer should reward the student (action close to +1). If the trainer's action isn't part of its reward, it receives the same reward even if it punishes the student (action close to -1), requiring the trainer to explore extensively via trial and error to understand the impact of its actions. To help the trainer better understand how its actions impact the reward it receives, we define the reward function such that it multiplies the scaled discriminator's output by trainer's actions. Therefore, the trainer agent's reward function is defined as $R^T = \upsilon(D_\phi(s^T))(a^T)$, where $D_\phi(s^T)$ is the output of the discriminator and $\upsilon(x) = 2x - 1$ is the scaling function. By incorporating $a^T$ into the reward function, the trainer learns to adjust its policy based on the effectiveness of its previous actions. The optimization problem of the trainer can be defined as

$$\max_{\pi_T} \mathbb{E}_{\substack{(s,a) \sim \pi_S \\ a^T \sim \pi_T}} [\upsilon(D_\phi(s,a))a^T]. \tag{7}$$

**RILe**  RILe combines the three components defined previously in order to find a student policy that mimics expert behaviors presented in $\tau_E$. In RILe, the student policy $\pi_S$ and the trainer policy $\pi_T$ can be trained via any single-agent online reinforcement learning method. The training algorithm is given in Appendix J. Overall, the student agent aims to recover the optimal policy $\pi_S^*$ defined as

$$\pi_S^* = \arg\max_{\pi_S} \mathbb{E}_{(s^S, a^S) \sim \pi_S} \left[ \sum_{t=0}^{\infty} \gamma^t [\pi_T((s_t^S, a_t^S))] \right]. \tag{8}$$

At the same time, the trainer agent aims to recover the optimal policy $\pi_T^*$ as

$$\pi_T^* = \arg\max_{\pi_T} \mathbb{E}_{\substack{s^T \sim \pi_S \\ a^T \sim \pi_T}} \left[ \sum_{t=0}^{\infty} \gamma^t [\upsilon(D_\phi(s_t^T))a_t^T] \right]. \tag{9}$$

We outline the employed training strategies in Appendix C.

## 5 EXPERIMENTS

We evaluate the performance of RILe by addressing three key questions:

1. How does RILe's adaptive reward function evolve compared to baseline methods and how does this evolution enhance the learning process?
2. How dynamic is RILe's adaptive reward function, and how does this adaptability benefit the student agent compared to the policy-agnostic discriminator in AIL?
3. Is RILe efficient and scalable to high-dimensional continuous control tasks?
4. Can RILe use expert-data explicitly to imitate expert behavior?

To answer the first question, we compare RILe's performance with AI(R)L baselines in a maze setting, where we demonstrate how the trainer agent modifies the reward function to guide the student during training. For the second question, we evaluate the dynamics of the learned reward function and analyze the correlation between these changes and improvements in the student's performance. For the third question, we evaluate RILe's effectiveness in imitating motion-capture data within robotic control tasks, using LocoMujoco (Al-Hafez et al., 2023), and imitating expert demonstrations in standard tasks, using (Brockman et al., 2016; Todorov et al., 2012). To answer the last question, we use a humanoid character from MuJoCo (Brockman et al., 2016; Todorov et al., 2012) to evaluate RILe's performance when expert data is explicitly used by the agents. Additional experimental results are provided in the Appendix, where we evaluate the robustness of the learned reward function and analyze the noise resilience of our method.

**Baselines** We compare RILe with seven baseline methods: Behavioral cloning (BC (Bain & Sammut, 1995; Ross & Bagnell, 2010), BCO (Torabi et al., 2018a)), adversarial imitation learning (GAIL (Ho & Ermon, 2016), GAIfO (Torabi et al., 2018b) and DRAIL (Lai et al., 2024)), adversarial inverse reinforcement learning (AIRL (Fu et al., 2018)), and inverse reinforcement learning (IQ-Learn (Garg et al., 2021)). DRAIL (Lai et al., 2024) introduces a diffusion-based discriminator implementation, which is applied to both GAIL and RILe, and referred as DRAIL-GAIL and DRAIL-RILe.

Additional experimental details are provided in the Appendix D, and hyperparameter selections are discussed in the Appendix H.

### 5.1 EVOLVING REWARD FUNCTION

To evaluate the impact of RILe's trainer agent on the learning process in an interpretable manner, we designed a maze experiment. Using a single expert demonstration, we trained RILe, GAIL, and AIRL, in a maze where the agent must navigate from a fixed start to a goal, avoiding obstacles.

Fig. 3 shows how each method's reward function evolves during training. For RILe, we plot the reward function learned by the trainer. For GAIL and AIRL, we visualize the discriminator outputs. The columns represent reward landscapes at 25%, 50%, 75%, and 100% of training completion. The student's trajectory from the previous epoch is overlaid to demonstrate how reward functions adapt to the student's progress.

RILe's reward function dynamically adapts to the student's current policy, providing meaningful guidance even when the discriminator easily distinguish non-expert policies. In contrast, although GAIL and AIRL's reward functions converge faster, they remain relatively static and lack RILe's adaptability, which is essential in more complex tasks. RILe's dynamic adaptation creates a learning curriculum that encourages exploration and gradual improvement toward expert-like behavior.

Specifically, the first column shows RILe's trainer encourage exploration towards the expert path when the student does not resemble the expert, which shows the trainer provides informative rewards

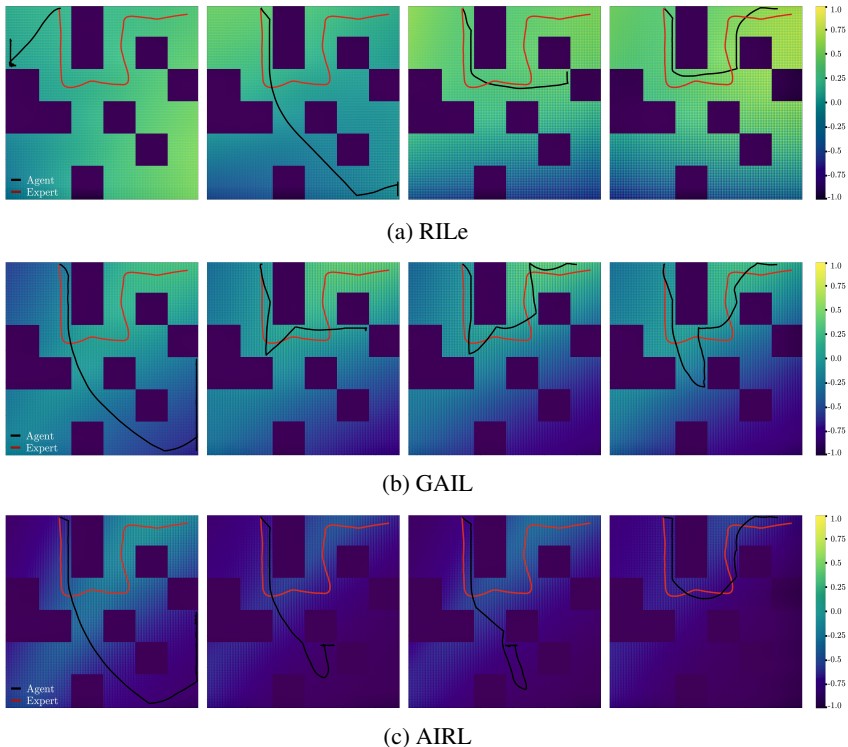

(a) RILe

(b) GAIL

(c) AIRL

Figure 3: **Reward Function Comparison**. Evolution of reward functions during training for (a) RILe, (b) GAIL, and (c) AIRL in a continuous maze environment. Columns show reward landscapes at 25%, 50%, 75%, and 100% of training completion (left to right). The expert's trajectory is shown in red, while the student agent's trajectory from the previous training epoch is in black. Color gradients represent reward values, with darker colors indicating lower rewards and brighter colors indicating higher rewards. Purple squares represent obstacles. RILe demonstrates a more adaptive reward function that evolves with the student's progress, while GAIL and AIRL maintain relatively static reward landscapes throughout training.

despite negative discriminator feedback. The second column presents when the student learns to reach the bottom-right, the trainer shifts high rewards to the top-left, guiding the agent to explore that area. Third column shows as the student approaches the goal, the trainer increases rewards around it while maintaining rewards in specific areas (e.g., the left part) to prevent the agent from getting stuck.

All in all, RILe's evolving reward function demonstrates its ability to provide meaningful guidance even when the discriminator easily identifies non-expert policies. By adapting to the student's current capabilities, RILe creates a dynamic learning curriculum that encourages exploration and gradual improvement towards expert-like behavior.

## 5.2 REWARD FUNCTION DYNAMICS

To understand the dynamics of the learned reward functions, we evaluated the adaptability of the reward functions and analyzed the correlation between the changes in the reward function and improvements in the student's performance. We compared RILe with GAIL, DRAIL-GAIL, and DRAIL-RILe in a task of learning to walk with the UnitreeH1 robot in LocoMujoco.

We introduced three metrics (see D.2 for more details): Reward Function Distribution Change (RFDC), Fixed-State Reward Function Distribution Change (FS-RFDC), and Correlation between Performance and Reward (CPR). RFDC measures the Wasserstein distance between reward distributions over consecutive training intervals, quantifying the overall shift in the reward function. FS-RFDC assesses how reward values for a fixed set of expert states change over time, where fixed states are all states present in the expert demonstration. CPR assesses how the performance improvement in the student agent is related to the updates in the reward function.

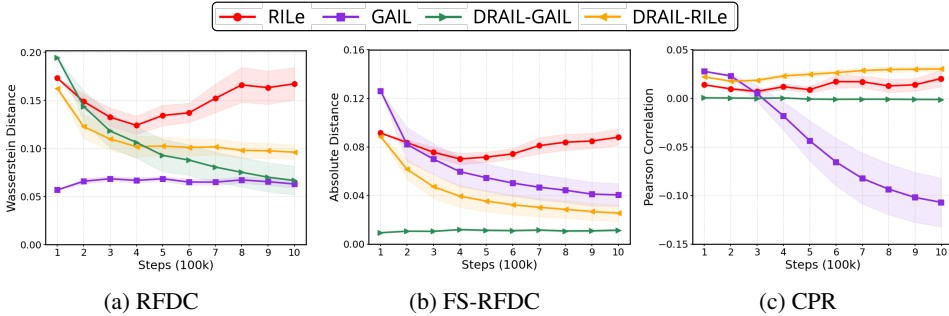

(a) RFDC  (b) FS-RFDC  (c) CPR

Figure 4: **Dynamics of Reward Functions**. **(a) Reward Function Distribution Change (RFDC):** Wasserstein distance between reward function distributions. **(b) Fixed-State Reward Function Distribution Change (FS-RFDC):** Mean absolute deviation of reward values for a fixed set of expert states. **(c) Correlation between Performance and Reward (CPR):** Pearson correlation between changes in the reward function and changes in the student's performance.

### 5.2.1 ADAPTABILITY OF THE LEARNED REWARD FUNCTION

We assess how dynamic the reward function learned by the trainer is compared to that of AIL. Fig. 4a presents changes in reward distributions over 10,000 consecutive steps. RILe exhibits the highest adaptability in its reward function, aligning with our goal of having the reward function adapt based on the student's learning stage. The advanced discriminator in DRAIL reduces the need for drastic reward function changes, yet RILe remains more adaptive than GAIL. Additionally, Fig. 4b shows deviations in reward values for the fixed set of states. Again, RILe's reward function is the most adaptive among all methods.

### 5.2.2 CORRELATION BETWEEN THE LEARNED REWARD AND THE STUDENT PERFORMANCE

We evaluate how changes in the reward function correlate with improvements in student performance. To this end, Fig. 4c presents the Pearson correlation between student's performance and reward updates. DRAIL-RILe achieves the highest positive correlation, indicating that it learns the most effective rewards for improving student performance. RILe ranks second, demonstrating that the trainer agent effectively helps the student achieve better scores. In contrast, for GAIL, the correlation starts positive but quickly becomes negative, which persists throughout training. This highlights the limitations of the policy-agnostic discriminator in effectively guiding the student.

## 5.3 MOTION-CAPTURE DATA IMITATION FOR ROBOTIC CONTINUOUS CONTROL

We evaluate RILe's performance on the LocoMujoco benchmark, which presents a challenging task of imitating motion-capture data for various robotic control tasks. This benchmark is particularly demanding due to its high dimensionality and the absence of action data in the motion-capture recordings which precludes the use of methods such as BC that require complete state-action pairs.

Table 1: Test results on seven LocoMujoco tasks.

| | | RILe | GAIL | AIRL | IQ | BCO | GAIfO | DRAIL GAIL | DRAIL RILe | Expert |
|---|---|---|---|---|---|---|---|---|---|---|
| **Walk** | Atlas | 870.6 | 792.7 | 300.5 | 30.9 | 21.0 | 834.2 | 834.4 | **899.1** | 1000 |
| | Talos | 842.5 | 442.3 | 102.1 | 4.5 | 11.9 | 710.0 | 787.7 | **896.6** | 1000 |
| | UnitreeH1 | 966.2 | 950.2 | 568.1 | 8.8 | 34.8 | 526.8 | 940.8 | **995.8** | 1000 |
| | Humanoid | **831.3** | 181.4 | 80.1 | 4.5 | 3.5 | 706.5 | 814.6 | 527.6 | 1000 |
| **Carry** | Atlas | **850.8** | 669.3 | 256.4 | 36.8 | 20.3 | 810.1 | 516.6 | 317.1 | 1000 |
| | Talos | 220.1 | 186.3 | 134.2 | 10.5 | 10.3 | 212.5 | 836.7 | **840.5** | 1000 |
| | UnitreeH1 | 788.3 | 634.6 | 130.5 | 14.4 | 21.1 | 604.5 | 796.7 | **909.5** | 1000 |

Table 1 presents the results for seven LocoMujoco tasks across test seeds (see D.3 for more details). RILe demonstrates superior performance in all scenarios, particularly excelling in generalization to new initial conditions as evidenced by the test seed results.

## 5.4 LEARNING FROM DEMONSTRATIONS

We evaluate RILe's performance on four Mu-JoCo tasks (see D.4 for more details), where baselines have been previously evaluated. Table 2 presents RILe effectively learns to perform close to or better than baselines.

Table 2: Test results on four MuJoCo tasks.

|  | RILe | GAIL | AIRL | IQ |
|---|---|---|---|---|
| Humanoid | **5928** | 5709 | 5623 | 327 |
| Walker2d | 4435 | **4906** | 4823 | 270 |
| Hopper | **3417** | 3361 | 3014 | 310 |
| HalfCheetah | **5205** | 4173 | 3991 | 755 |

## 5.5 IMPACT OF EXPERT DATA ON TRAINER-STUDENT DYNAMICS

We study how explicitly incorporating expert data into RILe's training affects the trainer's ability to adapt to the student's needs, in MuJoCo's *Humanoid* environment (Todorov et al., 2012; Brockman et al., 2016) using a single expert trajectory from (Garg et al., 2021). We varied the proportion of expert data in the replay buffers from 0% to 100%; for example, 25% means a quarter of the buffer is expert data and 75% is from the agent (see D.5 for more details).

Fig. 5 presents introducing the expert data led to faster convergence but decreased performance. Notably, when environmental interactions were completely replaced by expert data (100% case), the student's performance declined significantly. Excessive expert data appears to hinder the trainer's ability to adapt to the student, disrupting RILe's dynamic learning process. We include results from IQLearn and BC, which rely explicitly on expert data. Neither matches RILe's performance, even when RILe used substantial amounts of expert data.

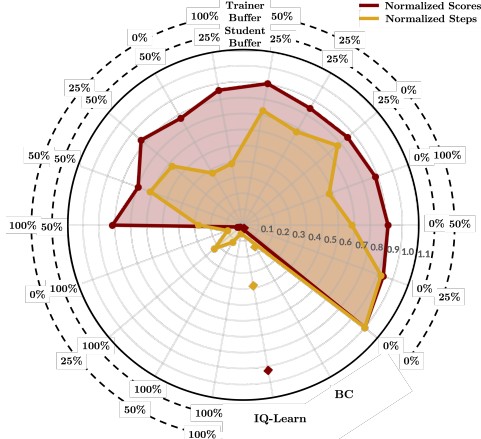

Figure 5: **Explicit Usage of Expert Data**. Red and yellow markers show normalized scores and steps, respectively. Expert data usage speeds the training of RILe but reduce final performance.

## 6 DISCUSSION

As our experiments demonstrate, RILe consistently outperforms baseline models across various settings thanks to its adaptive learning approach, where the trainer agent dynamically adjusts the reward function based on the student's current learning stage.

Our Maze experiments exemplify how the trainer agent adapts rewards based on the student's current training stage. The trainer encourages the student to take actions that are suboptimal in terms of immediate imitation but optimal for long-term learning. This adaptive strategy enables RILe to achieve better performance compared to baselines in our continuous control experiments. In contrast, as shown in Section 5.2, the policy-agnostic discriminators of AIL methods fail to provide constructive guidance in complex settings, limiting the student's improvement, limiting the student's ability to improve. Meanwhile, RILe's trainer continues to offer informative rewards, highlighting the importance of adaptive reward mechanisms.

However, RILe faces challenges in maintaining policy stability with a changing reward function. Freezing the trainer is effective but limits further adaptation, and the discriminator tends to overfit quickly. Future work could focus on exploring methods from fully cooperative multi-agent reinforcement learning to allow continuous adaptation, establishing bounds for trainer updates, and exploring discriminator-less approaches.

Despite these challenges, RILe demonstrates significant advantages in adaptability, robustness, and generalization. By providing dynamic and tailored rewards, it effectively guides the student through complex learning processes, making it a promising direction for future research in imitation learning and opening up new possibilities for dynamic and responsive learning frameworks.

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

# A  POMDP OF THE TRAINER

Partially Observable Markov Decision Process (POMDP) of the trainer is defined as $\text{POMDP}_T = (S_T, A_T, \Omega_T, T_T, O_T, R_T, \gamma)$. Here, $T_T = \{P(.\mid f^T, a^T)\}$ is the transition dynamics where $P(.\mid f^T, a^T)$ is the state distribution upon taking action $a \in A_T$ in state $f \in S_T$. The transition function incorporates the student's policy $\pi_S$, which evolves in response to the rewards provided, reflecting the hidden dynamics due to the unobserved $\pi_S$. The observation function $O_T = \{P(s^T \mid f^T, a^T)\}$ defines the probability of observing $s^T \in \Omega_T$ given the state $(f^T, a^T)$. The trainer deterministically observes the student's state-action pair, so $P(s^T = (s^S, a^S) \mid f^T, a^T) = 1$, where $f^T = (s^S, a^S, \pi_S)$.

# B  JUSTIFICATION OF RILE

**Assumptions:**

- The discriminator loss curve is complex and the discriminator function, $D_\phi(s, a)$, is sufficiently expressive since it is parameterized by a neural network with adequate capacity.
- For the trainer's and student's policy functions $(\pi^{\theta_T})$ and $(\pi^{\theta_S})$, and the Q-functions $(Q^{\theta_S})$, each is Lipschitz continuous with respect to its parameters with constants $(L_{\theta_T}), (L_{\theta_S}), and (L_Q)$, respectively. This means for all $(s, a)$ and for any pair of parameter settings $(\theta, \theta') : [|\pi^\theta(s, a) - \pi^{\theta'}(s, a)| \le L_\theta|\theta - \theta'|,][|Q^\theta(s, a) - Q^{\theta'}(s, a)| \le L_Q|\theta - \theta'|.]$

To prove that the student agent can learn expert-like behavior, we need to show that the trainer agent learns to give higher rewards to student experiences that match with the expert state-action pair distribution, as this would enable a student policy to eventually mimic expert behavior.

## B.1  LEMMA 1:

Given the discriminator $D_\phi$, the trainer agent optimizes its policy $\pi^{\theta_T}$ via policy gradients to provide rewards that guide the student agent to match expert's state-action distributions.

**Proof for Lemma 1** The student agent, $\pi_S(a_t^S|s_t^S)$, interacts with the environment and generates state-action pairs as $(s_t^S, a_t^S)$. The trainer agent observes these pairs and provides a reward $r_t^S = a_t^T = \pi_T(a_t^T|(s_t^S, a_t^S))$ to the student, where $a_t^T \in [-1, 1]$ is the trainer's action. We have $D_\phi : \mathcal{S} \times \mathcal{A} \to [0, 1]$ as the discriminator, parameterized by $\phi$, which outputs the likelihood that a given state-action pair $(s, a)$ originates from the expert, as opposed to the student.

The trainer's reward at timestep $t$ is:

$$r_t^T = \upsilon(D_\phi(s_t^T))a_t^T \tag{10}$$

where $s_t^T = (s_t^S, a_t^S)$ is the trainer's observation, $D_\phi(s_t^T)$ is the discrimantor output that estimates the likelihood that $s_t^T$ comes from the expert data, and $\upsilon(D) = 2D - 1$ is a scaling function that maps discriminator's output to the range $[-1, 1]$.

The trainer maximizes the expected cumulative reward:

$$J_T(\pi_T) = \mathbb{E}_{\pi_T, \pi_S}\left[\sum_{t=0}^{\infty} \gamma^t r_t^T\right] \tag{11}$$

where $\gamma \in [0, 1)$ is the discount factor. In other words, trainer aims to find the policy that maximizes $J_T(\pi_T)$: $\pi^{*T} = \arg\max_{\pi^T} J_T(\pi_T)$.

From the policy gradient theorem, the gradient of the trainer's objective with respect to the policy parameters, $\theta_T$, is:

$$\nabla_{\theta_T} J_T(\pi_T) = \mathbb{E}_{\pi_T, \pi_S}\left[\nabla_{\theta_T} \log \pi_T(a_t^T|s_t^T)Q_T(s_t^T, a_t^T)\right] \tag{12}$$

where $Q_T(s_t^T, a_t^T)$ is the action-value function of the trainer. The action-value function, $Q_T(s_t^T, a_t^T)$, and the value function, $V_T(s_t^T)$ is defined by Bellman equation as:

$$Q_T(s_t^T, a_t^T) = r_t^T + \gamma\mathbb{E}_{s_{t+1}^T}\left[V_T(s_{t+1}^T)\right] \tag{13}$$

$$V_T(s_{t+1}^T) = \mathbb{E}_{a_t^T \sim \pi_T}\left[Q_T(s_t^T, a_t^T))\right] \tag{14}$$

The trainer aims to maximize $Q_T(s_t^T, a_t^T)$ to satisfy Equation 12. Since $r_t^T$ depends directly on $D_\phi(s_t^T)$ and $a_t^T$, the trainer learns to select $a_t^T$ that maximizes $Q_T(s_t^T, a_t^T)$. Considering that $a_t^T \in [-1, 1]$, the immediate reward $r_t^T$ is maximized when $a_t^T$ has the same sign as $\upsilon(D_\phi(s_t^T))$. Therefore, the optimal action $a_t^{*T}$ is:

$$\alpha_t^{*T} = \begin{cases} 1, & \text{if } D\phi(s_t^T) > 0.5, \\ -1, & \text{if } D\phi(s_t^T) < 0.5, \\ \text{any value in } [-1, 1], & \text{if } D\phi(s_t^T) = 0.5. \end{cases} \quad (15)$$

Equation 15 implies the trainer assigns positive rewards to student state-action pairs that the discriminator assesses as more likely to be from the expert ($D_\phi(s_t^T) > 0.5$) and negative rewards to those unlikely to be from the expert ($D_\phi(s_t^T) < 0.5$). By this mechanism, the trainer's policy optimization relies on the discriminator's assessment to assign rewards that encourage expert-like behavior.

All in all, the derivative of the trainer's expected reward, Equation 12, with respect to its policy parameters is rewritten as:

$$\nabla_{\theta_T} J_T(\pi_T) = \mathbb{E}_{\pi_T, \pi_S}\left[\nabla_{\theta_T} \log \pi_T(a_t^T | s_t^T)\left((2D_\phi(s_t^T) - 1)a_t^T + \gamma Q_T(s_{t+1}^T, a_{t+1}^T)\right)\right] \quad (16)$$

The trainer adjusts $\pi_T$ to output high rewards when $D_\phi(s_t^T)$ is high. Therefore the trainer learns to assign higher rewards to student behaviors that are more similar to the expert behaviors, according to the discriminator.

### B.2 LEMMA 2:

The discriminator $D_\phi$, parameterized by $\phi$ will converge to a function that estimates the probability of a state-action pair being generated by the expert policy, when trained on samples generated by both a student policy $\pi^{\theta_S}$ and an expert policy $\pi_E$.

**Proof for Lemma 2**: The discriminator's objective is to distinguish between state-action pairs generated by the expert and those generated by the student. The training objective for the discriminator is framed as a binary classification problem over expert demonstrations and student-generated trajectories. The discriminator's loss function $\mathcal{L}_D(\phi)$ is the binary cross-entropy loss, which is defined as:

$$L_D(\phi) = -\mathbb{E}_{(s,a)\sim p_E}[\log(D_\phi(s, a))] - \mathbb{E}_{(s,a)\sim p_{\pi_S}}[\log(1 - D_\phi(s, a))]. \quad (17)$$

where $p_E(s, a)$ is the state-action distribution of the expert policy, and $p_{\pi_S}(s, a)$ is the state-action distribution of the student agent. Considering that $x = (s, a)$, this loss can be rewritten as:

$$L_D(\phi) = -\int [p_E(s, a) \log D_\phi(s, a) + p_{\pi_S}(s, a) \log(1 - D_\phi(s, a))] \, ds \, da \quad (18)$$

$$L_D(\phi) = -\int [p_E(x) \log D_\phi(x) + p_{\pi_S}(x) \log(1 - D_\phi(x))] \, dx. \quad (19)$$

As presented in Goodfellow et al. (2014), the optimal discriminator that minimizes this loss, $D_\phi^*$, is:

$$D_\phi^*(x) = \frac{p_E(x)}{p_E(x) + p_{\pi_S}(x)}, \quad (20)$$

$$D_\phi^*(s, a) = \frac{p_E(s, a)}{p_E(s, a) + p_{\pi_S}(s, a)}. \quad (21)$$

This shows that the optimal discriminator estimates the probability that a state-action pair comes from the expert policy, normalized by the total probability from both expert and student policies.

# C   TRAINING STRATEGIES

The introduction of the trainer agent into the AIL framework introduces instabilities that can hinder the learning process. To address these challenges, we employ three strategies.

**Freezing the Trainer Agent Midway:** Continuing to train the trainer agent throughout the entire process can lead to overfitting on minor fluctuations in the student's behavior. This overfitting causes the trainer to assign inappropriate negative rewards, which diverts the student away from expert behavior—especially since the student agent may fail to interpret these subtle nuances correctly in the later stages of training. To prevent this, we freeze the trainer agent once its critic network within the actor-critic framework converges during the training process.

We consider the trainer's critic network to have converged when the change in the exponential moving average (with a smoothing factor of 0.99) of the critic output and its variance over a window of 50000 training iterations fall below a certain threshold. In all our experiments, this threshold is set to 0.1, which we found empirically after our hyperparameter search (see Appendix H). This threshold works for all settings where the reward is bounded between $-1$ and $1$, which is the case for all our experiments.

**Reducing the Trainer's Target Network Update Frequency:** We decrease the target network update frequency of the trainer agent to half that of the student agent. After our hyperparameter sweeps (see Appendix H), we empirically found that updating at half the student agent's frequency works best. This adjustment aims to prevent overestimation bias in the trainer's value function and to slow down its learning pace. By updating less frequently, the trainer provides more consistent and reliable reward signals. This steadier guidance helps the student agent better understand and adapt to the trainer's rewards, facilitating more stable learning.

**Increasing the Student Agent's Exploration:** We increase the exploration rate of the student agent compared to standard AIL methods. We implement an epsilon-greedy strategy within the actor-critic framework, allowing the student to occasionally take random actions. This increased exploration enables the student to visit a wider range of state-action pairs. Consequently, the trainer agent receives diverse input, helping it learn a more effective reward function. This diversity is crucial for the trainer to observe the outcomes of various actions and to guide the student more effectively toward expert behavior.

# D   EXPERIMENTAL SETTINGS

## D.1   EVOLVING REWARD FUNCTION

We use single expert demonstration in this experiment. For RILe, we plot the reward function learned by the trainer. For GAIL, we visualize the discriminator output, and for AIRL, the reward term under the discriminator.

## D.2   REWARD FUNCTION DYNAMICS

In this experiment, we select the student agent's hyperparameters to be identical to those used in GAIL, ensuring that the only difference between the agents is the reward function. Therefore, we use the best hyperparameters identified for GAIL, applied to both GAIL and RILe, from our hyperparameter sweeps in Appendix H.

**RFDC:** We calculate the Wasserstein distance between reward distributions over consecutive 10,000-step training intervals, denoted as times $t$ and $t + 10,000$. This metric quantifies how much the overall reward distribution shifts over time. Changes in reward distributions depend both on the reward function and the student policy updates. Since we use the same student agent with the same hyperparameters, higher RFDC values still indicate that the reward function is adapting more dynamically in response to the student's learning progress.

**FS-RFDC:** We compute the mean absolute deviation of rewards between consecutive 10,000-step training intervals for a fixed set of states derived from expert data. As the fixed set, we use all the states in the expert data. Since the states used for calculating rewards are fixed, changes in this value

purely depend on the reward function updates. This metric assesses how the reward values for specific states change over time.

**CPR:** We evaluate how changes in the reward function correlate with improvements in student performance. We store rewards from both the learned reward function and the environment-defined rewards in separate buffers. In other words, we collect samples from two reward functions: the learned reward function and the environment-defined reward function. The environment rewards consider the agent's velocity and stability. Every 10,000 steps, we calculate the Pearson correlation between these rewards and empty the buffers. This metric evaluates whether increases in the learned rewards relate to performance enhancements.

### D.3 MOTION-CAPTURE DATA IMITATION FOR ROBOTIC CONTINUOUS CONTROL

During training, we use 8 different random seeds and 8 distinct initial positions for the robot. The validation setting mirrors the training conditions: we sample initial positions from the same set of 8 possibilities and use the same random seeds. In this setting, the student agent selects actions deterministically, allowing us to assess its performance under familiar conditions.

For the test setting, we evaluate the policy's ability to generalize to new, unseen scenarios. We modify the initial positions of the robot by randomly initializing it in stable configurations not included in the fixed set used during training. Additionally, we use different random seeds from those in training, introducing new random variations that affect the environment's dynamics during state transitions. This setup enables us to assess how well the learned policy performs when faced with novel initial conditions and environmental changes.

### D.4 LEARNING FROM DEMONSTRATIONS

Each method is trained using 25 expert trajectories provided in the IQ-Learn paper Garg et al. (2021). We use single seed for the training, and after the training, run experiments with 10 different random seeds and report the mean and standard deviation of the results.

### D.5 IMPACT OF EXPERT DATA ON TRAINER-STUDENT DYNAMICS

In this experiment, both seeds and initial positions in the test setting are different from the training one, and we report values from the test setting.

For every percentage of the expert-data in buffers, we continue trainings of both the trainer agent and the student agent of RILe. For instance, in 100% expert data in the trainer's buffer case, both the student and the discriminator are trained normally using samples from the student agent. However, we didn't include student's state-action pairs to the trainer's buffer, instead, we filled that buffer with a batch of expert data, and updated the trainer regularly using this modified buffer. Similarly, in 100% expert data in the student's buffer case , we trained the trainer agent and the discriminator normally, using samples from the student agent. However, student's state-actions pairs are not included in the student's buffer, and student agent is updated just by using expert state-action pairs, using rewards coming from the trainer agent for these expert pairs.

Regarding the normalizations, we trained Behavioral Cloning (BC) and RILe across various data leakage levels, selecting the highest-scoring run (0% leakage RILe) as the baseline. Other scores and convergence steps are normalized by dividing by the score and convergence steps of the baseline (0% leakage RILe). For IQLearn, we used their reported numbers in their paper, as we couldn't replicate their results with their code and hyperparameters.

## E ADDITIONAL EXPERIMENTS

### E.1 NOISY EXPERT DATA

To demonstrate the advantage of using RL to learn the reward function in RILe, as opposed to deriving the reward directly from the discriminator in AIL and AIRL, we designed a 5x5 MiniGrid experiment. The grid consists of 4 lava tiles that immediately kill the agent if it steps in it, representing terminal conditions. The goal condition of the environment is reaching the green tile.

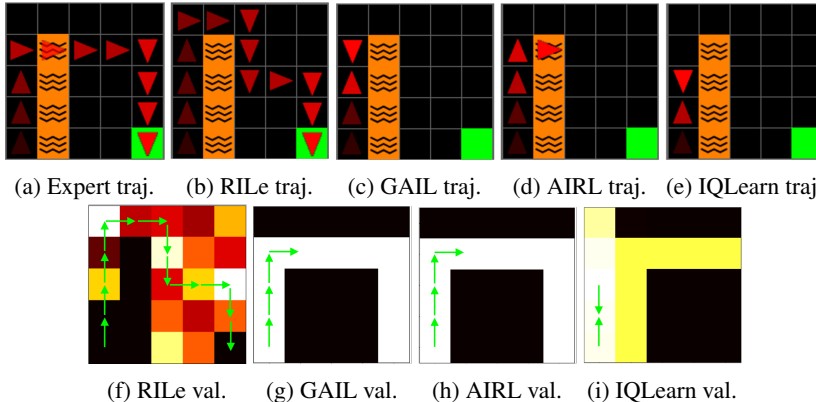

(a) Expert traj.    (b) RILe traj.    (c) GAIL traj.    (d) AIRL traj.    (e) IQLearn traj.

(f) RILe val.    (g) GAIL val.    (h) AIRL val.    (i) IQLearn val.

Figure 6: In a 5x5 grid environment with lava, (a) the expert trajectory is characterized by noisy data that passes through lava without resulting in death. (c) GAIL, (d) AIRL and (e) IQLearn learn to imitate the expert's path precisely, leading them to either get stuck near the lava or enter it and perish. (b) RILe avoids the noisy data, better mimics the expert in later stages, and successfully reaches the goal. Subfigures (f-i) display the value tables for RILe, GAIL, AIRL, and IQLearn respectively. The optimal path, derived from the reward of the trainer or discriminator, is highlighted with green lines.

The expert demonstrations are imperfect, depicting an expert that passes through a lava tile without being killed and still reaches the green goal tile. Using this data, we trained the adversarial approaches with a perfect discriminator, which provides a reward of $0.99$ if the visited state-action pair stems from the expert and $0.01$ otherwise. These values were chosen over 1 and 0 because both AIRL and GAIL use the logarithm of the discriminator output to calculate rewards.

Results are presented in Fig. 6. The value graphs (Fig. 6e-g) are attained by computing the value of each grid cell $c_i$ as $\sum_{a \in A} D(c_i, a)$ for AIRL and GAIL, and $\sum_{a \in A} \pi_T(c_i, a)$ for RILe. Fig. 6a shows the expert trajectory.

GAIL (Fig. 6c), AIRL (Fig. 6d) and IQLearn (Fig. 6e) fail to reach the goal, as their agents either become stuck or are directed into lava.

In contrast, RILe (Fig. 6d) successfully reaches the goal, demonstrating its ability to navigate around imperfections in expert data. The difference in the value graphs between RILe and the baselines intuitively explains this outcome. In AIL and AIRL (Fig. 6f-g), the optimal paths, defined by the actions most rewarded by their discriminators, follow the noisy expert data perfectly. Similarly, in IQLearn, the agent tries to match expert state-actions as closely as possible, minimizing any deviation from the expert trajectory. In contrast, RILe's trainer agent, trained using RL, adds an extra degree of freedom in the adversarial IL/IRL setting. By providing rewards that maximize cumulative returns from the discriminator, rather than deriving the reward directly from its output, the value graph (Fig. 6f) can learn to circumvent the lava tile in order to follow the expert trajectory to the goal. Consequently, the optimal path of the student agent can overcome the sub-optimal state suggested by the noisy expert demonstration. Since the student agent is guided by the trainer to also match the expert trajectory, it remains close to this path after passing the lava tiles.

### E.2 ROBUSTNESS TO NOISE IN THE EXPERT DATA

To evaluate the robustness of RILe and baseline methods to noise in the expert data, we conducted experiments in the MuJoCo Humanoid-v2 environment. Artificial noise sampled from a zero-mean Gaussian distribution with varying standard deviations ($\Sigma$) was added to a single expert trajectory, affecting either the actions or the states. The baselines used for comparison were GAIL (Ho & Ermon, 2016), AIRL (Fu et al., 2018), RIL-Co (Tangkaratt et al., 2021), IC-GAIL (Wu et al., 2019), and IQ-Learn (Garg et al., 2021).

As shown in Table 3, RILe consistently outperforms the baselines across different noise levels, demonstrating superior robustness even when a high amount of noise is present in the expert data ($\Sigma = $

Table 3: Test results in MuJoCo Humanoid-v2 environment, where artificial noise sampled from a zero-mean Gaussian distribution is added to a single expert trajectory. Results are aggregated over 20 different-seed environments. IQ-Learn* is trained using the official code and hyperparameters of the IQ-Learn algorithm.

| | Noise-Free | Action Noise | | State Noise | |
|---|---|---|---|---|---|
| | $\Sigma = 0$ | $\Sigma = 0.2$ | $\Sigma = 0.5$ | $\Sigma = 0.2$ | $\Sigma = 0.5$ |
| RILe | **5681** | **5280** | **5154** | **5350** | **5205** |
| GAIL | 5430 | 5275 | 902 | 5147 | 917 |
| AIRL | 5276 | 4869 | 4589 | 4898 | 4780 |
| RIL-Co | 576 | 491 | 493 | 505 | 501 |
| IC-GAIL | 610 | 601 | 568 | 590 | 591 |
| IQ-Learn* | 312 | 192 | 153 | 243 | 277 |

0.5). These results indicate that RILe is less sensitive to imperfections in the expert demonstrations compared to existing methods.

### E.3    ROBUSTNESS OF THE LEARNED REWARD FUNCTION

We evaluated the robustness of the reward functions learned by RILe and AIRL (Fu et al., 2018) through an experiment similar to that conducted by Xu et al. (2022). Initially, both methods were trained to learn reward functions in a noise-free MuJoCo Humanoid-v2 environment. After training, these reward functions were frozen. Subsequently, new student agents were trained using these fixed reward functions in environments where Gaussian noise was added to the agents' actions, with varying noise levels.

Table 4 presents the results of this evaluation. The reward function learned by RILe demonstrates superior robustness to noise, maintaining high performance even under increased noise levels. In contrast, the performance of agents using the reward function learned by AIRL decreases more significantly as noise increases. These findings indicate that the reward function learned by RILe is more resilient to environmental noise, contributing to better agent performance in noisy conditions.

Table 4: We test the robustness of learned reward functions. After training reward functions in a noise-free setting, reward functions are frozen, and used to train a new agent in a noisy environment, where Gaussian noise is added to agent's actions in every step.

| | No Noise | Mild Noise | High Noise |
|---|---|---|---|
| | $\Sigma = 0$ | $\Sigma = 0.2$ | $\Sigma = 0.5$ |
| RILe | **5748** | **5201** | **5196** |
| AIRL | 5334 | 5005 | 4967 |

### E.4    REWARD CURVES

We compare the reward curves of RILe, GAIL (Ho & Ermon, 2016), AIRL (Fu et al., 2018), IQ-Learn (Garg et al., 2021), and AdapMen (Liu et al., 2023) in the MuJoCo Humanoid-v2 experiment. Since the task involves learning from expert trajectories, we combined AdapMen with an adversarial discriminator to enable training without an expert policy.

As shown in the reward curves, despite RILe having multiple components, it is the most efficient method. This efficiency is achieved through the dynamic guidance of the trainer during training, which adapts the reward function to meet the student's needs.

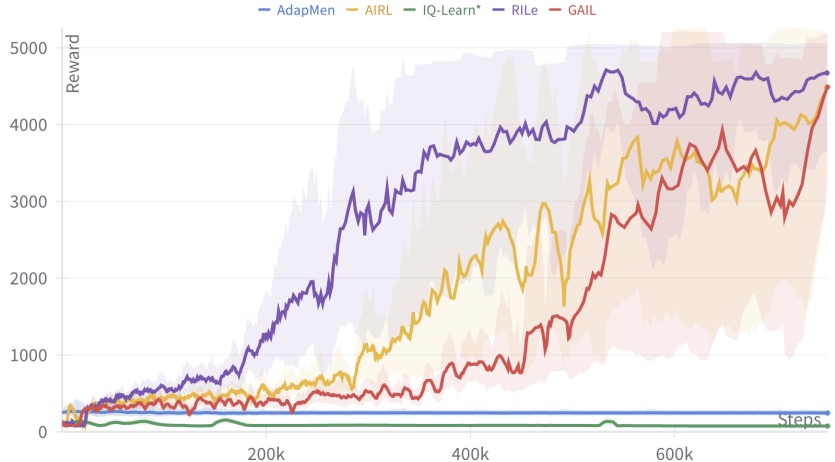

Figure 7: Training reward curves for the MuJoCo Humanoid-v2 experiment comparing RILe, AIRL, GAIL, IQ-Learn*, and adapted AdapMen. AdapMen is combined with an adversarial discriminator to be able to train it without expert policy.

## F  EXTENDED MUJOCO RESULTS

We present MuJoCo results for the test setting, with standard errors, in Table 5.

Table 5: Test results on four MuJoCo tasks with standard errors.

|              | RILe           | GAIL          | AIRL         | IQLearn       | DRAIL        |
|--------------|----------------|---------------|--------------|---------------|--------------|
| Humanoid-v2  | **5928 ± 188** | 5709 ± 63     | 5623 ± 252   | 327 ± 105     | 5755 ± 34    |
| Walker2d-v2  | 4435 ± 206     | **4906 ± 159** | 4823 ± 221  | 270 ± 43      | 4016 ± 127   |
| Hopper-v2    | **3417 ± 155** | 3361 ± 51     | 3014 ± 190   | 310 ± 47      | 1230 ± 73    |
| HalfCheetah-v2 | **5205 ± 31** | 4173 ± 94    | 3991 ± 126   | 755 ± 211     | 4133 ± 41    |

## G  EXTENDED LOCOMUJOCO RESULTS

We present LocoMujoco results for the validation setting and test setting, with standard errors, in Table 6 and 7, respectively.

Table 6: Validation results on seven LocoMujoco tasks.

| | | RILe | GAIL | AIRL | IQ | BCO | GAIfO | DRAIL GAIL | DRAIL RILe | Expert |
|---|---|---|---|---|---|---|---|---|---|---|
| Walk | Atlas | 895.4 ±25 | 918.6 ±133 | 356.0 ±68 | 32.1 ±4 | 28.7 ±4 | 831.6 ±41 | 741.3 ±46 | 773.9 ±13 | 1000 |
| | Talos | 884.7 ±8 | 675.5 ±105 | 103.4 ±22 | 7.2 ±2 | 19.9 ±4 | 718.8 ±16 | 963.7 ±48 | 949.4 ±54 | 1000 |
| | UnitreeH1 | 980.7 ±15 | 965.1 ±20 | 716.2 ±124 | 12.5 ±6 | 43.7 ±8.4 | 586.6 ±102 | 954.7 ±20 | 973.5 ±8 | 1000 |
| | Humanoid | 970.3 ±101 | 216.2 ±18 | 78.2 ±6 | 6.8 ±1 | 8.3 ±1 | 345.7 ±34 | 550.8 ±148 | 595.3 ±73 | 1000 |
| Carry | Atlas | 889.7 ±44 | 974.2 ±80 | 271.9 ±30 | 39.5 ±8 | 42.7 ±9 | 306.2 ±9 | 654.1 ±109 | 344.1 ±28 | 1000 |
| | Talos | 503.3 ±72 | 338.5 ±48 | 74.1 ±8 | 11.7 ±3 | 8.1 ±1 | 444.5 ±96 | 889.8 ±163 | 874.3 ±174 | 1000 |
| | UnitreeH1 | 850.6 ±80 | 637.4 ±90 | 140.9 ±21 | 12.3 ±2 | 30.2 ±5 | 503.6 ±55 | 620.8 ±60 | 878.1 ±46 | 1000 |

Table 7: Test results on seven LocoMujoco tasks.

| | | RILe | GAIL | AIRL | IQ | BCO | GAIfO | DRAIL GAIL | DRAIL RILe | Expert |
|---|---|---|---|---|---|---|---|---|---|---|
| **Walk** | Atlas | 870.6 ±13 | 792.7 ±105 | 300.5 ±74 | 30.9 ±10 | 21.0 ±3 | 803.1 ±68 | 834.4 ±23 | **899.1** ±17 | 1000 |
| | Talos | 842.5 ±24 | 442.3 ±76 | 102.1 ±17 | 4.5 ±3 | 11.9 ±1 | 687.2 ±44 | 787.7 ±11 | **896.6** ±12 | 1000 |
| | UnitreeH1 | 966.2 ±14 | 950.2 ±13 | 568.1 ±156 | 8.8 ±3 | 34.8 ±10 | 526.8 ±72 | 940.8 ±20 | **995.8** ±6 | 1000 |
| | Humanoid | **831.3** ±98 | 181.4 ±24 | 80.1 ±9 | 4.5 ±2 | 3.5 ±2 | 292.1 ±25 | 814.6 ±80 | 527.6 ±39 | 1000 |
| **Carry** | Atlas | **850.8** ±62 | 669.3 ±55 | 256.4 ±47 | 36.8 ±14 | 20.3 ±1 | 402.9 ±39 | 516.6 ±60 | 317.1 ±19 | 1000 |
| | Talos | 220.1 ±88 | 186.3 ±28 | 134.2 ±18 | 10.5 ±3 | 10.3 ±2 | 212.5 ±32 | 836.7 ±160 | **840.5** ±133 | 1000 |
| | UnitreeH1 | 788.3 ±71 | 634.6 ±45 | 130.5 ±22 | 14.4 ±2 | 21.1 ±6 | 504.5 ±30 | 796.7 ±131 | **909.5** ±9 | 1000 |

## H   HYPERPARAMETERS

We present hyperparameters in Table 8. For DRAIL, we replaced the discriminators with the implementation provided by DRAIL and adopted their hyperparameters for the HandRotate task.

Our experiments revealed that RILe's performance is particularly sensitive to certain hyperparameters. We highlight three key observations:

- RILe is more sensitive to the hyperparameters of the discriminator compared to other methods. Specifically, increasing the discriminator's capacity or training speed, by using a larger network architecture or increasing the number of updates per iteration, adversely affects RILe's performance. A powerful discriminator tends to overfit quickly to the expert data, resulting in high confidence when distinguishing between expert and student behaviors. This poses challenges for the trainer agent, as the discriminator's feedback becomes less informative.

- The update frequency of the trainer agent's target network influences the stability of the RILe framework. Lower update frequencies lead to improved stability. A slower-updating trainer provides more consistent reward signals, allowing the student agent to better adapt to the rewards. However, a lower update frequency slows down the learning process, as the trainer adapts more slowly to changes in the student's behavior. Therefore, there is a trade-off between stability and learning speed that needs to be balanced.

- Enhancing the exploration rate of the student agent benefits RILe more than it does baseline methods. By encouraging the student to explore more, through strategies like higher entropy regularization or implementing an epsilon-greedy policy, the student visits a broader range of state-action pairs. This increased diversity provides the trainer agent with more varied data, enabling it to learn a more effective and robust reward function. The additional exploration helps the trainer to better capture the effects of different actions.

## I   COMPUTE RESOURCES

For the training of RILe and baselines, following computational sources are employed:

- AMD EPYC 7742 64-Core Processor
- 1 x Nvidia A100 GPU
- 32GB Memory

Table 8: Hyperparameter Sweeps and Best Hyperparameters for LocoMujoco and Humanoid Experiments

| | Hyperparameters | RILe | GAIL | AIRL | IQ-Learn |
|---|---|---|---|---|---|
| **Discriminator** | Updates per Round | **1**, 2, 8 | 1, 2, 8 | 1, 2, 8 | - |
| | Batch Size | **32**, 64, 128 | **32**, 64, 128 | **32**, 64, 128 | - |
| | Buffer Size | 8192, **16384**, 1e5 | 8192, **16384**, 1e5 | 8192, **16384**, 1e5 | - |
| | Network | [512FC, 512FC] [256FC, 256FC] [**64FC, 64FC**] | [512FC, 512FC] [256FC, 256FC] [**64FC, 64FC**] | [512FC, 512FC] [256FC, 256FC] [**64FC, 64FC**] | - |
| | Gradient Penalty | 0.5, **1** | 0.5, **1** | 0.5, **1** | - |
| | Learning Rate | 3e-4, 1e-4, **3e-5**, 1e-5 | 3e-4, 1e-4, **3e-5**, 1e-5 | 3e-4, 1e-4, **3e-5**, 1e-5 | - |
| **Student** | Buffer Size | 1e5, **1e6** | 1e5, **1e6** | 1e5, **1e6** | 1e5, **1e6** |
| | Batch Size | 32, **256** | 32, **256** | 32, **256** | 32, **256** |
| | Network | [**256FC, 256FC**] | [**256FC, 256FC**] | [**256FC, 256FC**] | [**256FC, 256FC**] |
| | Activation Function | **ReLU**, Tanh | **ReLU**, Tanh | **ReLU**, Tanh | **ReLU**, Tanh |
| | Discount Factor ($\gamma$) | **0.99**, 0.97, 0.95 | **0.99**, 0.97, 0.95 | **0.99**, 0.97, 0.95 | **0.99**, 0.97, 0.95 |
| | Learning Rate | **3e-4**, 1e-4, 3e-5, 1e-5 | **3e-4**, 1e-4, 3e-5, 1e-5 | **3e-4**, 1e-4, 3e-5, 1e-5 | **3e-4**, 1e-4, 3e-5, 1e-5 |
| | Tau ($\tau$) | 0.05, **0.01**, 0.005 | 0.05, **0.01**, 0.005 | 0.05, **0.01**, 0.005 | 0.05, **0.01**, 0.005 |
| | Epsilon-greedy | 0, 0.1, **0.2** | **0**, 0.1, 0.2 | **0**, 0.1, 0.2 | **0**, 0.1, 0.2 |
| | Entropy | **0.2**, 0.5, 1 | **0.2**, 0.5, 1 | **0.2**, 0.5, 1 | 0.05, 0.1, **0.2**, 0.5, 1 |
| **Trainer** | Buffer Size | 8192, **16384**, 1e5, 1e6 | - | - | - |
| | Batch Size | 32, **256** | - | - | - |
| | Network | [**256FC, 256FC**] [64FC, 64FC] | - | - | - |
| | Activation Function | **ReLU**, Tanh | - | - | - |
| | Discount Factor ($\gamma$) | **0.99**, 0.97, 0.95 | - | - | - |
| | Learning Rate | **3e-4**, 1e-4, 3e-5, 1e-5 | - | - | - |
| | Tau ($\tau$) | 0.05, 0.01, **0.005** | - | - | - |
| | Entropy | **0.2**, 0.5, 1 | - | - | - |
| | Freeze Threshold | 1, 0.5, **0.1**, 0.01, 0.001 | - | - | - |

## J ALGORITHM

---

**Algorithm 1** RILe Training Process

---

1: Initialize student policy $\pi_S$ and trainer policy $\pi_T$ with random weights, and the discriminator $D$ with random weights.
2: Initialize an empty replay buffer $B$
3: **for** each iteration **do**
4:      Sample trajectory $\tau_S$ using current student policy $\pi_S$
5:      Store $\tau_S$ in replay buffer $B$
6:      **for** each transition $(s, a)$ in $\tau_S$ **do**
7:          Calculate student reward $R^S$ using trainer policy:

$$R^S \leftarrow \pi_T \tag{22}$$

8:          Update $\pi_S$ using policy gradient with reward $R^S$
9:      **end for**
10:    Sample a batch of transitions from $B$
11:    Train discriminator $D$ to classify student and expert transitions

$$\max_D E_{\pi_S}[\log(D(s,a))] + E_{\pi_E}[\log(1 - D(s,a))] \tag{23}$$

12:      **for** each transition $(s, a)$ in $\tau_S$ **do**
13:          Calculate trainer reward $R^T$ using discriminator:

$$R^T \leftarrow \upsilon(D(s,a))a^T \tag{24}$$

14:          Update $\pi_T$ using policy gradient with reward $R^T$
15:      **end for**
16: **end for**

---

---

**Algorithm 2** RILe Training Process with Off-policy RL

---

1: Initialize student policy $\pi_S$, trainer policy $\pi_T$, and the discriminator $D$ with random weights.
2: Initialize an empty replay buffers $B_D$, $B_S$, $B_T$ with different sizes
3: **for** each iteration **do**
4:      Sample trajectory $\tau_S$ using current student policy $\pi_S$
5:      Store $\tau_S$ in replay buffers $B_D$, $B_S$, $B_T$ a batch of transitions, $b_S$ from $B_S$
6:      **for** each transition $(s, a)$ in $b_S$ **do**
7:          Calculate student reward $R^S$ using trainer policy:

$$R^S \leftarrow \pi_T \tag{25}$$

8:          Update $\pi_S$ using calculated rewards
9:      **end for**
10:    Sample a batch of transitions $b_D$ from $B_D$
11:    Train discriminator $D$ to classify student and expert transitions

$$\max_D E_{\pi_S}[\log(D(s,a))] + E_{\pi_E}[\log(1 - D(s,a))] \tag{26}$$

12:    Sample a batch of transitions, $b_T$ from $B_T$
13:      **for** each transition $(s, a)$ in $b_T$ **do**
14:          Calculate trainer reward $R^T$ using discriminator:

$$R^T \leftarrow \upsilon(D(s,a))a^T \tag{27}$$

15:          Update $\pi_T$ using calculated rewards
16:      **end for**
17: **end for**

---

