# OpenReview forum: "RILe: Reinforced Imitation Learning"
_ICLR.cc/2025/Conference — Submitted to ICLR 2025_

### Official Review · Reviewer_RivH · 2024-10-16

**Soundness:** 2
**Presentation:** 2
**Contribution:** 2
**Rating:** 5
**Confidence:** 4

**Summary:**

The paper proposed Reinforced Imitation Learning (RILe) to learn the reward function for policy learning with AIL and teacher-student framework. The method is evaluated on LocoMujoco benchmark and it outperforms GAIL and AIRL.

**Strengths:**

- The idea of Decoupled Reward-function Learning with AIL is novel.

- RILe demonstrates better performance on the LocoMujoco benchmark than baselines (GAIL and AIRL).

**Weaknesses:**

- The connection between motivation and the proposed method is weak. More intuition or theoretical proofs are needed.
- While the paper highlights some issues with existing methods, it lacks detailed references and experiments to convincingly demonstrate how RILe addresses these issues. More ablation studies or theoretical analysis could strengthen the argument.
- The scope of the experiment is limited to comparing only GAIL and AIRL on one benchmark. More baselines [1,2,3,4] and tasks should be evaluated.

**Questions:**

- In line 241, the authors mention "the trainer’s RL-based approach optimizes for long-term performance rather than immediate feedback".  If the benefit comes from RL, all AIL methods also share it since they all use RL for policy learning. How does the RILe handle this?

- In Section 5.1 and Figure 3, it’s stated that RILe offers better rewards for guiding student policies, but in the second and third columns, GAIL appears to learn faster. The reward contours for GAIL and AIRL also seem closer to the ground truth. Moreover, learning with a fixed reward function is the most common approach in typical RL. Could the authors elaborate more on how the evolving feature benefits policy learning?

- Why did the authors choose LocoMujoco, which lacks action labels, rather than more widely used Mujoco tasks for Learning from Demonstration (LfD)? I would recommend limiting the scope to Learning from Demonstration (LfD), where action labels are available. Besides, IQLearn did provide learning with state-only rewards version in the paper. If the authors want to demonstrate the application on Learning from Observation (LfO), I think it will be better to conduct experiments separately with a different set of baselines [5,6,7,8].

- In Figure 4, how are normalized scores and steps defined? The result seems to be applicable to many online imitation learning methods. Could the authors provide more details on the purpose of these metrics in the context of RILe?

- The authors mentioned RILe improves the computational efficiency compared to other IRL works. However, RILe needs to train an extra trainer network compared to GAIL. Is there a quantitative analysis on this?

References:

[1] Pomerleau, D. A. (1991). Efficient training of artificial neural networks for autonomous navigation. Neural computation, 3(1), 88-97.

[2] Papagiannis, G., & Li, Y. (2022, September). Imitation learning with sinkhorn distances. In Joint European Conference on Machine Learning and Knowledge Discovery in Databases (pp. 116-131). Cham: Springer Nature Switzerland.

[3] Chi, C., Xu, Z., Feng, S., Cousineau, E., Du, Y., Burchfiel, B., ... & Song, S. (2023). Diffusion policy: Visuomotor policy learning via action diffusion. The International Journal of Robotics Research, 02783649241273668.

[4] Lai, C. M., Wang, H. C., Hsieh, P. C., Wang, Y. C. F., Chen, M. H., & Sun, S. H. (2024). Diffusion-Reward Adversarial Imitation Learning. arXiv preprint arXiv:2405.16194.

[5] Torabi, F., Warnell, G., & Stone, P. (2018). Behavioral cloning from observation. arXiv preprint arXiv:1805.01954.

[6] Liu, M., Zhu, Z., Zhuang, Y., Zhang, W., Hao, J., Yu, Y., & Wang, J. (2022). Plan your target and learn your skills: Transferable state-only imitation learning via decoupled policy optimization. arXiv preprint arXiv:2203.02214.

[7] Torabi, F., Warnell, G., & Stone, P. (2018). Generative adversarial imitation from observation. arXiv preprint arXiv:1807.06158.

[8] Huang, B. R., Yang, C. K., Lai, C. M., Wu, D. J., & Sun, S. H. (2024). Diffusion Imitation from Observation. arXiv preprint arXiv:2410.05429.

---

> ### Author Response · Authors · 2024-12-03
> **Response to Reviewer RivH**
>
> We appreciate the reviewer’s active engagement in the discussion and we thank the reviewer for their constructive suggestions during the rebuttal.
>
> We understand the concerns about the performance of the IQ-Learn. For the comparison with IQ-Learn, we used the official codebase, configurations, and data provided by the authors. Unfortunately, the authors have explicitly stated that they are unable to maintain the framework; and suggested using their pretrained policies, but these are not available for certain environments like Humanoid and Walker. Given this limitation, we adhered strictly to the available resources and guidelines. In the camera-ready version, we will also report the scores achieved by their pretrained policies for Hopper and HalfCheetah:
>
> **Hopper-v2:** 2698.36 &pm; 18.37
> **HalfCheetah-v2:** 5093.94 &pm; 29.7
>
> We believe this represents the best possible approach under the circumstances.
>
>
> Regarding the empirical results, we compare RILe with **eight state-of-the-art imitation learning baselines.** RILe consistently outperformed these baselines  across the following four experiments:
> 1. **Imitating robotic locomotion skills from motion-capture data (Section 5.3)**, compared with GAIL, AIRL, DRAIL, IQ-Learn, GAIfO, BCO.
> 2. **Imitating continuous control skills in a default reinforcement learning benchmark (Section 5.4)**, compared with GAIL, AIRL, IQ-Learn, DRAIL (results included in the appendix and will be in the camera-ready version).
> 3. **Learning from noisy expert data (Appendix E.2)**, compared with GAIL, AIRL, IQ-Learn, RIL-Co, IC-GAIL.
> 4. **Acquiring robust reward functions  (Appendix E.3)**, compared with AIRL.
>
> Additionally, we have three intuitive experiments to illustrate the distinct learning dynamics of RILe compared to traditional adversarial imitation learning methods:
>
> 1. **Evolving Reward Function  (Section 5.1):** Illustrating how RILe works different than baselines, by presenting the evolution of the learned reward function
> 2. **Reward Function Dynamics (Section 5.2):** Quantitatively analyzing the adaptability of RILe’s reward function and the effects of the reward function on the student performance
> 3. **Noisy Expert Data  (Appendix E.1):**  Visually presenting how RILe works different to baselines when trained  with noisy expert data (Appendix E.1)
>
> While more theoretical analysis could further strengthen our work, we believe that the comprehensive empirical evidence provided, spanning multiple tasks and baselines, strongly supports the contribution of RILe.

---

### Official Review · Reviewer_D3w2 · 2024-10-31

**Soundness:** 3
**Presentation:** 2
**Contribution:** 3
**Rating:** 6
**Confidence:** 3

**Summary:**

RILe (Reinforced Imitation Learning environment) is a framework that enhances reinforcement and imitation learning by introducing a dynamic, adaptive reward function responsive to agent performance. Addressing limitations of traditional IL methods like GAIL and AIRL, RILe’s trainer-student model continuously optimizes rewards through reinforcement learning, effectively guiding the student agent to mimic expert demonstrations in complex tasks. Experiments in maze and humanoid locomotion tasks demonstrate RILe's advantages in scalability, robustness, and adaptability, highlighting its effectiveness for high-dimensional control environments.

**Strengths:**

1. **Motivation and Intuition**: The motivation for reducing reliance on static discriminators is compelling, as complex tasks often necessitate adaptive guidance.

2. **Novelty**: The framework’s use of a trainer agent that dynamically learns a reward function to align with student policy is an innovative approach that enhances traditional adversarial learning strategies.

3. **Technical Contribution**: RILe’s decoupling of the reward function learning from the student’s policy learning shows clear benefit, especially in tasks requiring extensive exploration, as the framework enables more progressive feedback than binary classifications.

4. **Clarity**: The paper is well-structured, with clear theoretical explanations and effective visualizations. Figures, such as Figure 3, successfully illustrate the evolution of reward functions across training stages.

**Weaknesses:**

1. **Limited Baselines** (important): The experiment lacks more recent baselines (BC from 2010, GAIL from 2016, and AIRL from 2018). It would strengthen the work to include comparisons with more recent AIL methods, such as DRAIL [1], V-MAIL [2], or ARC [3]. Especially, they should cite DRAIL [1], which also focus on improving learning efficiency, particularly in complex, high-dimensional environments like RILe does.

2. **Lack of Ablation Study**: The paper would benefit from additional ablation studies to better analyze the contributions of individual components within the RILe framework.

3. **Limited Task Diversity**: RILe is tested only on LocoMujoco (locomotion tasks). Expanding experiments to include navigation or manipulation tasks would improve the generalizability of the framework.

[1] Lai, Chun-Mao, et al. "Diffusion-Reward Adversarial Imitation Learning." arXiv preprint arXiv:2405.16194 (2024).

[2] Rafailov, Rafael, et al. "Visual adversarial imitation learning using variational models." Advances in Neural Information Processing Systems 34 (2021): 3016-3028.

[3] Deka, Ankur, Changliu Liu, and Katia P. Sycara. "ARC-Actor Residual Critic for Adversarial Imitation Learning." Conference on Robot Learning. PMLR, 2023.

**Questions:**

1. The experimental setup relies primarily on older baselines (e.g., BC from 2010, GAIL from 2016, and AIRL from 2018). Including comparisons with more recent adversarial imitation learning (AIL) methods, such as DRAIL [1], would provide a more comprehensive assessment of RILe’s performance, especially given that DRAIL also address challenges related to learning efficiency in complex, high-dimensional environments. The paper needs to include DRAIL [1]. Doing so would significantly enhance the paper’s impact. I would be willing to score it higher with this addition.

2. How does the dynamic reward adjustment by the trainer agent handle rapidly changing or highly stochastic environments, where optimal behavior may vary significantly within episodes?

3. RILe involves multiple components with different objectives (trainer, student, and discriminator). How sensitive is RILe to hyperparameter choices across these agents, and are there any recommended tuning strategies?

4.  The paper shows RILe’s ability to perform well with noisy expert data, but how robust is this approach if the noise distribution changes dynamically during training, or if the expert data is sparse?

[1] Lai, Chun-Mao, et al. "Diffusion-Reward Adversarial Imitation Learning." arXiv preprint arXiv:2405.16194 (2024).

---

### Official Review · Reviewer_BWR3 · 2024-11-04

**Soundness:** 1
**Presentation:** 2
**Contribution:** 2
**Rating:** 3
**Confidence:** 3

**Summary:**

This paper proposes a novel method for adversarial imitation learning. Unlike established methods such as GAIL and AIRL, which learn a policy using reinforcement learning (RL) under a reward function derived directly from the discriminator, the proposed method, RILe, introduces an additional RL component called the trainer agent. This trainer agent generates the reward function for policy learning, effectively serving as an intermediary layer between the discriminator and the policy learning component (referred to as the student agent). The paper claims that this additional trainer component, as an RL agent, is able to dynamically provide an adaptive reward function throughout the student agent's policy learning process, resulting in superior imitation learning performance in complex settings.

**Strengths:**

1. This paper aims to address some of the challenges faced by existing imitation learning methods in complex settings, a valuable problem to study with a wide range of potential practical applications.

2. The introduction of an additional trainer agent in the proposed method, RILe, is a novel design.

3. The discussion of related literature is clear and effectively situates the proposed method within the context of existing research.

**Weaknesses:**

1. My major concern with the paper is that several major claims regarding the capabilities of the proposed method, RILe, are not sufficiently supported by rigorous analysis or adequate experimental evidence.
* Claim of RILe's adaptability: For instance, in Section 5.1, it is stated that "RILe demonstrates a more adaptive reward function that evolves with the student's progress ...". However, this claim is inadequately supported by the limited results in Figure 3, which only includes a single, simple setting and is open to subjective interpretation. Similar claims in the Introduction and Discussion suggest that RILe’s adaptability leads to more efficient policy learning for the student agent, but there is insufficient experimental evidence to validate this connection.
* Claim of RILe outperforming state-of-the-art adversarial IL methods: Although the main results in Section 5.2 show that RILe outperforms the baseline methods, GAIL and AIRL, on several LocoMujuco tasks, these results lack sufficient details on the experimental setup, such as the settings of tasks used for evaluation and the hyperparameter values of the compared methods.
* Additionally, the proof of the theoretical argument, Lemma 1, is neither rigorously presented nor complete.

2. Another concern I have with the proposed RILe method is that it stacks one RL agent (the trainer agent) on top of another (the student agent). The trainer agent's state transition depends on the student agent's policy since its states are represented by the state-action pairs generated by the student agent. Meanwhile, the student agent's policy is influenced by the trainer agent's policy through the reward function generation. This structure creates a dependency in the trainer's RL environment (i.e. the state transition of the RL environment faced by the trainer agent is not exogonous to its policy), making the MDP ill-defined and potentially introducing extra instability to the training framework.
* In fact, the paper briefly addresses training instability in the Discussion section and mentions a workaround of freezing the trainer midway through training.

3. The proposed optimization objective of the trainer agent, Equation 8, lacks a clear explanation of the rationale behind this particular choice. (See Question 1 below)

4. Overall, the paper is not well-organized and includes errors or inaccuracies in some parts.
* In Section 3.2, entropy regularization is introduced as part of the RL problem formulation in this work, defined in Equation 1. However, in Section 3.3, the RL formulation is presented without entropy regularization, which is then added again in Equation 2.
* In Section 4, the first term on the right-hand side of Equation 14 should be $v(D_{\phi}(s_t))a_t$ instead of $D_{\phi}(s_t)$ to remain consistent with the reward definition in Equation 11.
* The workaround for the training instability (freezing the trainer after meeting certain criteria) is only mentioned in the Discussion section, with no mention of it in the discussion of experimental settings and results.

**Questions:**

1. What is the rationale behind the specific definition of the optimization objective of the trainer agent, $v(D_{\phi}(s, a)) a^T$ (Equation 8)? It is mentioned that "By incorporating $a^T$ into the reward function, the trainer learns to adjust its policy based on the effectiveness of its previous actions", but this explanation is somewhat unclear. Could you clarify why $a^T$ is specifically multiplied with $v(D_{\phi}(s, a))$?

2. In Figure 4, the performance of RILe remains relatively good when the trainer's replay buffer is 100\% expert data, provided that the proportion of expert data in the student's replay buffer is not too large. This outcome is somewhat puzzling, as I would expect that with 100\% expert data in the trainer's replay buffer, neither the trainer nor the discriminator would be trained on any trajectories generated by the student policy, based on the framework illustrated in Figure 2. Am I misunderstanding something here?

---

> ### Author Response · Authors · 2024-11-21
> **Response to Reviewer BWR3**
>
> We thank the reviewer for their time and constructive suggestions.
>
> **W1.** We appreciate the reviewer for this constructive feedback. To further support our claims about the dynamic reward function, we include two more ablation studies:
> * Evaluating the dynamic adaptability of the reward function in RILe and baseline methods
> * Analyzing how the rewards provided by the trainer agent correlate with improvements in the student's performance.
>
> Regarding the concern about the experimental setting, we are including a new subsection under Appendix C, where we explain settings of tasks and the difference between test and training. Also, we include hyperparameter sweeps in the Appendix G. We also update the proof of Lemma 1 in Appendix A.1.
>
> **W2.** We solve instabilities introduced by the training agent through three different strategies: 1. Freezing the trainer agent once its actor-critic critic network converges, 2. Decreasing the target network update frequency of the trainer agent, 3. Increasing the exploration of the student agent. We are including a new section, Appendix B of the updated paper, where we explain the intuition behind them.
>
> **W3/Q1.** The trainer agent guides the student via taking actions between -1 and 1, and the discriminator output is bounded between 0 and 1. Imagine a case where student agent pair is (s*,a*), and let’s assume this pair is very similar to the ones in expert data, and the discriminator output for this pair is 1. In this case, we expect the trainer to reward the student, by taking an action close to 1. However, if reward is just defined without including trainer’s action (such as D_\phi(s,a)), the trainer cannot understand the effect of its’ own actions: in the imagined case, even if the trainer punishes the student by taking an action closer to -1, the reward of the trainer would be 1 if reward function is independent from the trainer’s action. Therefore, we incorporate the actions of the trainer inside its own reward function, such that it gives positive rewards to the trainer whenever the trainer rewards the student for expert-like behavior and punishes the student for non-expert-like behavior. We prefer multiplication in order to encourage teachers to take larger rewards around 1 and -1, since we observed the trainer agent takes small actions if we use subtracting.  The rationale behind selecting v(x)=2x−1 is in order to map the discriminator's output to the same ranges of the trainer.
>
> **W4.** We are updating the RL formulation in Section 3.3. to include the entropy term. We updated the proof of Lemma 1, including Equation 11-14. Regarding the workaround, we are including a new section under the appendix where we discuss the training strategies we use (Appendix B in the updated paper).
>
> **Q2.** We understand the reviewer’s concern and have provided a clearer explanation in Appendix C of the updated paper. In Section 5.3, we examined the impact of using modified training buffers by continuing the training process with different data compositions. Specifically, when the trainer’s buffer contained 100% expert data, we trained both the student agent and the discriminator normally using the student’s state-action pairs. However, we excluded the student’s state-action pairs from the trainer’s buffer and instead populated it exclusively with expert data batches, updating the trainer regularly with this modified buffer. This approach reduced the ability of the trainer to adapt to the current stage of the student, and decreased the performance by 20%. This highlights  the importance of the trainer that can adapt itself to support the student’s learning process.

---

> ### Author Response · Authors · 2024-12-04
> **Response to Reviewer BWR3**
>
> We appreciate the reviewer’s questions and concerns.
>
> The example in Section 5.1 **visually** demonstrates how reward functions change in response to a student's progress **in an interpretable manner.** While such visual demonstrations are valuable for intuitive understanding, we acknowledge that achieving this level of interpretability in more complex settings is challenging. Therefore, in complex tasks, we validate the benefit of our method and the adaptability of the trainer through the quantitative analyses presented in Section 5.2. The experiments in Section 5.3 demonstrate that standard methods without RILe’s adaptability fail to achieve similar performance. In the camera-ready version, we will emphasize more that this experiment in Section 5.1 is an intuitive analysis intended to present the learned reward functions alongside the student’s progress, which is supported by the quantitative results in Section 5.2.
>
> Figure 4(c) illustrates **how the adaptability of the reward function, as presented in Figures 4(a) and 4(b)**, affects the student's performance. In the case of a fixed reward function, the correlation with the environment reward in Figure 4(c) may be high, but the adaptability measured in Figures 4(a) and 4(b) would be zero. This lack of adaptability in fixed reward functions, typical in standard RL, necessitates careful reward engineering to account for different stages of training and exploration, which can be challenging. Therefore, analyzing Figure 4(c) in isolation does not provide a complete picture. By considering Figures 4(a), 4(b), and 4(c) together, we demonstrate that RILe’s learned reward function not only adapts more than baselines, but also positively impacts the student's performance.
>
> Regarding GAIL’s discriminator, we understand the confusion and appreciate the opportunity to clarify. The discriminator in GAIL is trained using isolated state-action pairs and focuses solely on distinguishing expert data from non-expert data. As training progresses, the discriminator becomes increasingly effective at this task, which may lead to penalization of state-action pairs that result in behavior similar to the expert but are not present in the expert data. Consequently, the discriminator does not guide the agent toward better performance if that performance arises from state-action pairs outside the expert demonstrations. This lack of consideration for the agent’s progress and adaptation results in the negative correlation between the environment reward (reflecting the student's performance) and the discriminator's reward. In contrast RILe's trainer agent adapts its rewards based on the student's progress to help the student imitate the expert. During training, the trainer explores different reward values to encourage behaviors leading toward expert performance, even if those behaviors involve state-action pairs not included in the expert data. The trainer even provides rewards for suboptimal actions that are steps toward expert behavior (as visually presented in Section 5.1). Even when the discriminator becomes highly effective at distinguishing expert data, the trainer continues to adapt and provides informative rewards that help the student align with the expert, which is more beneficial for the trainer in the long term in terms of the cumulative reward. This results in higher adaptability, a positive correlation with the environment reward, and better performance in complex tasks, as we present in Sections 5.3 and 5.4.
>
> As we stated in Section 4, *if the trainer’s reward depends only on the discriminator’s output, the trainer receives the same reward regardless of its action, offering no immediate feedback on the effectiveness of rewarding or penalizing the student, requiring the trainer to explore extensively via trial and error to understand the impact of its actions*. In our experiments, we found that while RILe still performs well with the naive discriminator reward, training time significantly increases. Specifically, with our current reward design (multiplying the discriminator score by the trainer's action), RILe converges in approximately 2 million steps on average across the seven tasks presented in Section 5.3. When using the naive discriminator reward without incorporating the trainer's action, the average steps until convergence increase to about 9 million, with similar end performance. This substantial difference demonstrates that our reward function design significantly improves training efficiency. We will include these sample-efficiency comparisons in the camera-ready version in the Appendix to provide clear comparison between two cases.

---

### Official Review · Reviewer_rwcm · 2024-11-05

**Soundness:** 3
**Presentation:** 3
**Contribution:** 3
**Rating:** 8
**Confidence:** 3

**Summary:**

This paper introduces ‘Reinforced Imitation Learning’, a new trainer-student framework. In this framework, the trainer receives feedback from a discriminator and continuously adjusts the output reward in response to the student's behavior. The student is then trained using reinforcement learning based on the reward signal from the trainer. This approach may enable increased exploration behavior in students and more detailed feedback from the trainer. The proposed approach is evaluated in a maze setting, LocoMujoco, and humanoid MuJoCo. Experimental results indicate that the proposed approach outperforms competitive baselines significantly.

**Strengths:**

**Originality & Significance**

1. The reviewer found the concepts of a custom reward function and the decoupling of reward function learning from the student and discriminator to be quite innovative. The experimental results suggest that this approach effectively addresses the issue of insufficient guidance, a common challenge in early-stage training for adversarial imitation and reinforcement learning methods. This improvement in guidance appears to be particularly beneficial during the initial phases of the learning process.

2. The background section is well-organized and effectively introduces the key concepts. The authors provide a concise overview of reinforcement learning (RL), inverse reinforcement learning (IRL), adversarial imitation learning (AIL), and adversarial reinforcement learning (AIRL). These explanations establish a solid foundation for understanding the proposed approach and its position within the broader literature. Additionally, the authors clearly articulate the distinctions between existing methods and their proposed approach, further highlighting the motivation and novelty of their work.


**Quality**

The claims are supported by experimental results. Particularly, table 1 shows that the proposed approach is able to handle complex tasks that baselines didn’t perform well.
The reviewer appreciated the presentation of figure 3. It clearly illustrates the dynamics of the reward function of the proposed approach and its difference to existing methods.

**Clarity**

While some clarification of  the technical contents is needed,  the paper is overall well organized and easy to follow. Please see the weakness section for details.

**Weaknesses:**

1. The reviewer found the reward function for the trainer agent needs further clarification. Specifically, L 241 reads ‘the trainer’s RL-based approach optimizes for long-term performance rather than immediate feedback’, but it is unclear to the reader how the formulation of the reward function (Eq. 8) could achieve it.

2. It is unclear to the reviewer the role of $a^T$ in the reward function of the trainer (Eq. 8). With $a^T$ in the objective, would it encourage the trainer to generate $a^T=1$ without further constraint? Also, L 314 reads ‘by incorporating $a^T$ into the reward function, the trainer learns to adjust its policy based on the effectiveness of its previous actions’. It is unclear to the reviewer how this is achieved. Please elaborate.

**Questions:**

1. The reviewer's primary concern is how the trainer's reward function contributes to optimizing long-term feedback. Additionally, the reviewer questions the rationale for incorporating the trainer's action into the reward function. Please refer to the "Weaknesses" section for a detailed explanation.
2. L313, how is the constant determined in $v(x) = 2x - 1$?
3. How many training environment steps are used for each task?
4. Why are confidence intervals not included in Table 1?

---

> ### Comment · Reviewer_rwcm · 2024-11-27
>
> I appreciate the authors' response, which addressed most of my concerns.

---

### Meta-Review · Area_Chair_jkky · 2024-12-21

**Metareview:**

This paper proposes a novel method for adversarial imitation learning with three components: a discriminator that provides feedback to a trainer (an RL agent) which is optimized to reward the student (another RL agent). The paper argues that the additional trainer component can adapt the reward function to accelerate the student agent's policy learning process, perhaps increasing exploration. Experiments demonstrate strong results in locomotion tasks.

Reviewers appreciated the novelty of the proposed approach and strong empirical performance (esp. during the early stages of learning).
Authors appreciated the clear discussion of the relationship with prior work and overall clear writing and motivation.

Several reviewers recommended additional baselines, ablation experiments, and evaluation tasks (e.g., manipulation). They also provided feedback on paper organization and suggested that the paper might be strengthened by including additional theoretical results. Reviewers also had questions about why the reward function (Eq. 8) resulted in maximizing for long term performance and details of the experimental setup. Reviewers also suggested that some of the claims should be clarified (e.g., is it the _adaptability_ that is driving improved performance?) and had concerns about the complexity of the proposed approach, as it seems to stack on RL agent on top of another. (this is briefly discussed in the paper's Discussion).

Overall, despite the novelty of the proposed approach, concerns about the experiments (i.e., whether they support the specific claims made in the paper) compel me to recommend that this paper not be accepted.

**Additional Comments On Reviewer Discussion:**

During the rebuttal, the authors clarified details of their method (e.g., why it optimizes for long-term rewards, how it avoids instabilities, and included many more details in Appendix C. During the rebuttal, reviewers noted that they found the empirical results insufficient, even though the core idea of the paper is novel. There were some last-minute posts by the reviewers which took a step towards addressing some of these concerns, but a follow-up discussion among reviewers clarified that concerns remained.

---

### Decision · Program_Chairs · 2025-01-22

Reject